# Giant tunnelling electroresistance in atomic-scale ferroelectric tunnel junctions

Yueyang Jia [1,7], Qianqian Yang [2,7], Yue-Wen Fang [3,4] ✉, Yue Lu [5], Maosong Xie [1], Jianyong Wei[1], Jianjun Tian [2], Linxing Zhang [2] ✉ & Rui Yang [1,6] ✉

Ferroelectric tunnel junctions are promising towards high-reliability and low-power non-volatile memories and computing devices. Yet it is challenging to maintain a high tunnelling electroresistance when the ferroelectric layer is thinned down towards atomic scale because of the ferroelectric structural instability and large depolarization field. Here we report ferroelectric tunnel junctions based on samarium-substituted layered bismuth oxide, which can maintain tunnelling electroresistance of $7 \times 10^5$ with the samarium-substituted bismuth oxide film down to one nanometer, three orders of magnitude higher than previous reports with such thickness, owing to efficient barrier modulation by the large ferroelectric polarization. These ferroelectric tunnel junctions demonstrate up to 32 resistance states without any write-verify technique, high endurance (over $5 \times 10^9$), high linearity of conductance modulation, and long retention time (10 years). Furthermore, tunnelling electroresistance over $10^9$ is achieved in ferroelectric tunnel junctions with 4.6-nanometer samarium-substituted bismuth oxide layer, which is higher than commercial flash memories. The results show high potential towards multi-level and reliable non-volatile memories.

The stable and electrically switchable spontaneous polarization in ferroelectric materials can lead to high-performance non-volatile memories[1–5]. Ferroelectric tunnel junctions (FTJs) with ultra-thin ferroelectric films have attracted much attention as two-terminal resistive switching devices due to the small cell size, non-destructive readout, and scalability towards large-scale arrays[6–11]. The ferroelectric thin film allows the quantum mechanical tunneling of electrons, with the average tunneling barrier and thus the tunneling electroresistance (TER) modulated by the polarization of the ferroelectric layer[12–14]. A high TER can increase the reliability and decrease the energy consumption in FTJ devices and arrays[9,15], and the barrier height or width modulation has been proven to improve the TER value[16]. By using a top metal electrode with a high work function, an increased average barrier height is obtained[17], which leads to a TER of about 100. Recently, barrier height modulation above 1 eV is demonstrated in Cr/CuIn$_2$S$_6$ (~4 nm)/graphene FTJs by tuning the graphene layer thickness[18], leading to a TER over $10^7$. Besides, it is also effective to modulate the effective barrier width to further enhance the TER. The niobium-doped strontium titanate (Nb:SrTiO$_3$, NSTO) has been used as a semiconducting bottom electrode to provide an extra Schottky-like barrier in the high-resistance state, owing to the depletion region at the ferroelectric/semiconductor

[1]University of Michigan—Shanghai Jiao Tong University Joint Institute, Shanghai Jiao Tong University, Shanghai 200240, China. [2]Beijing Advanced Innovation Center for Materials Genome Engineering, Institute for Advanced Materials and Technology, University of Science and Technology Beijing, Beijing 100083, China. [3]Fisika Aplikatua Saila, Gipuzkoako Ingeniaritza Eskola, University of the Basque Country (UPV/EHU), Europa Plaza 1, 20018 Donostia/San Sebastián, Spain. [4]Centro de Física de Materiales (CSIC-UPV/EHU), Manuel de Lardizabal Pasealekua 5, 20018 Donostia/San Sebastián, Spain. [5]Beijing Key Laboratory of Microstructure and Properties of Solids, Faculty of Materials and Manufacturing, Beijing, University of Technology, Beijing 100124, China. [6]State Key Laboratory of Radio Frequency Heterogeneous Integration, Shanghai Jiao Tong University, Shanghai 200240, China. [7]These authors contributed equally: Yueyang Jia, Qianqian Yang. ✉e-mail: yuewen.fang@ehu.eus; linxingzhang@ustb.edu.cn; rui.yang@sjtu.edu.cn

interface[19], and the TER over $10^6$ is achieved in a Pt/BaTiO$_3$ (BTO)/NSTO device[20].

Besides a high TER, the scaling of ferroelectric layer thickness is also critical for FTJs because it can reduce the program voltage and latency. However, as FTJs continue to scale down towards atomic scale, it is quite challenging to maintain a large TER when the ferroelectric layer thickness is down to the one-nanometer range, because the TER is limited by the structural instability of the ferroelectric phase and the enhanced depolarization field at nanometer-scale thickness[21,22]. For a 1-nm-BTO-based FTJ which is achieved using a large compressive strain imposed by a NdGaO$_3$ substrate[6], the TER is only about 2. FTJs based on 1-nm Hf$_{0.8}$Zr$_{0.2}$O$_2$ (HZO) with strain confinement show enhanced ferroelectricity[23], but the TER is still only about 200, far from that of commercial flash memories.

Recently, the strong out-of-plane ferroelectricity is demonstrated in layered bismuth oxide with samarium (Sm) substitution[24]. The samarium bonds stabilize the ferroelectricity even when the film is down to 1 nm, which shows strong potential for high-performance non-volatile memories. In this article, we demonstrate metal/ferroelectric/semiconductor FTJs based on Sm-substituted bismuth oxide (BSO), which can achieve a TER over $7 \times 10^5$ with a 1-nm ferroelectric BSO film. Such TER is more than three orders of magnitude higher than the TER values of previously reported FTJs with 1-nm ferroelectric layers. We also show that the TER can be further enhanced as the thickness of the BSO layer increases. The TER over $10^9$ has been obtained in FTJs with 4.6-nm-thick BSO films, which is among the highest TER reported in thin-film FTJs at room temperature. The giant TER is attributed to both ferroelectric barrier height modulation and the barrier height/width modulation at the surface depletion region of NSTO[19], thanks to the large polarization maintained at small thickness for the BSO layer[24]. These BSO-based FTJs can further support multi-level cells and analog memories. By varying the reset stop voltages, 32 distinct resistance states (5 bits of data storage) are realized in a single FTJ. Moreover, the devices demonstrate linear conductance modulation, more than $5 \times 10^9$ endurance cycles, and over 10 years of retention time. These BSO-based FTJs hold high promise towards reliable, high-performance, and low-power non-volatile memory and in-memory computing applications.

## Results

### Structure and ferroelectric properties of ultra-thin BSO films

Layered Sm-substituted bismuth oxide (Bi$_{2-x}$Sm$_x$O$_3$) is a recently discovered material with ferroelectric states driven by a lone pair of electrons, which can maintain a large polarization value down to 1 nm[24]. The large polarization at such a small thickness ($t$) is beneficial for high-performance FTJs. We have grown (001) epitaxial Bi$_{1.8}$Sm$_{0.2}$O$_3$ (BSO) thin films on NSTO substrates by a sol-gel method, followed by lithographically patterning the top electrodes (see Methods Section for details). The schematic diagram of the Au (60 nm)/Cr (15 nm)/BSO (1 nm)/NSTO device structure is shown in Fig. 1a, b. The atomic model of the BSO thin film is shown in Supplementary Fig. 1, and the atomic arrangement of the BSO film with thickness down to ~1 nm is confirmed by the high-angle annular dark-field scanning transmission electron microscopy (HAADF-STEM) (Fig. 1c). Due to the stress of the substrate and surface, the lattice of the ultrathin BSO film along the c axis is stretched to a certain extent, resulting in a stable polarization[25,26]. At the same time, X-ray reflectivity measurements and the fitting results confirm that the thickness of the BSO film is ~1 nm with a surface roughness of 0.173 nm[24], which is consistent with the theoretical model and the TEM image.

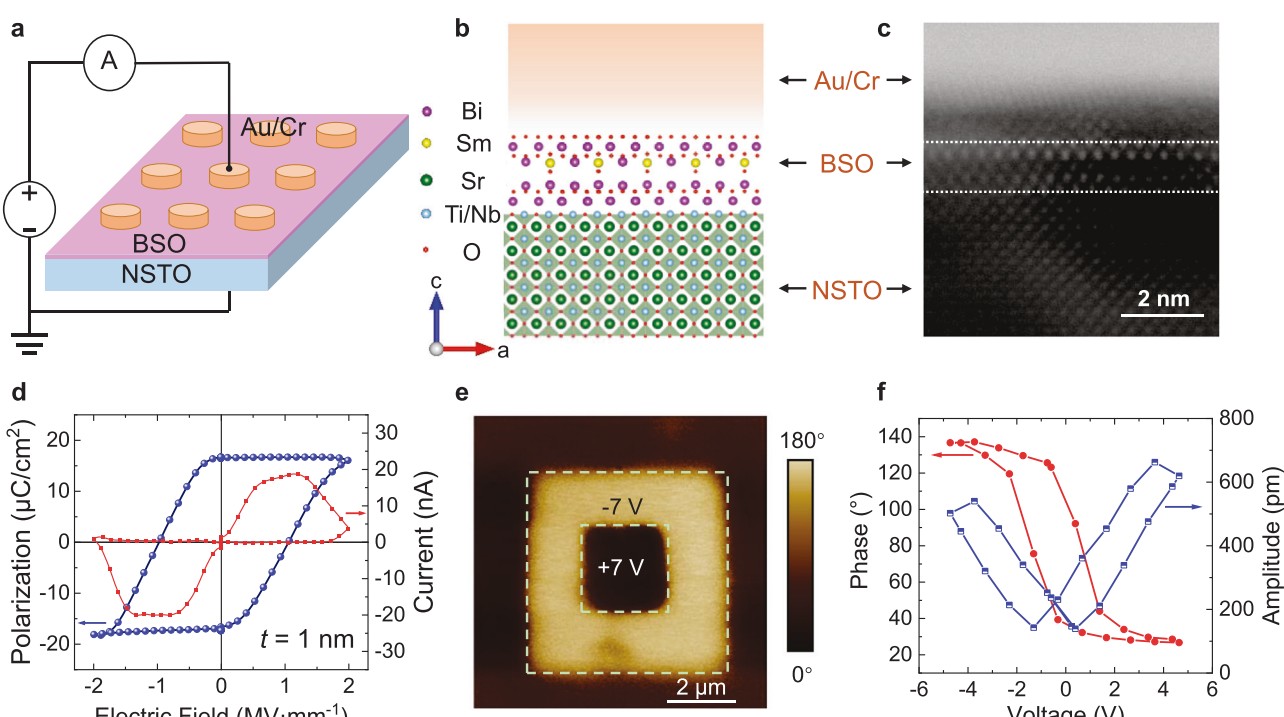

**Fig. 1 | Structure of the Au/Cr/BSO/NSTO-based FTJs and ferroelectric properties of the 1-nm BSO films. a** Schematic illustration of the FTJ structure. **b** Atomic-level diagram of the Au (60 nm)/Cr (15 nm)/BSO (1 nm)/NSTO FTJ structure. While the Sm atoms are distributed randomly in the structure, we provide one possible BSO structure obtained from a previous study[24]. **c** The HAADF-STEM image of a BSO film grown on NSTO substrate, with the BSO thickness of ~1 nm. **d** The polarization hysteresis loop (left axis) and switching current (right axis) of the 1-nm BSO, obtained by the PUND-mode measurement under an electric field at 625 Hz frequency. **e** The out-of-plane phase of the PFM measurement, shown in color scale after a box-in-box writing with a tip bias in the 1-nm-thick BSO film. The entire detection area is 8 μm × 8 μm. The tip biasing is noted in the corresponding area, and the region without the note is unbiased. **f** Off-field local PFM measurements for BSO films with the thickness of 1 nm, including both phase (left axis) and amplitude (right axis) hysteresis loops.

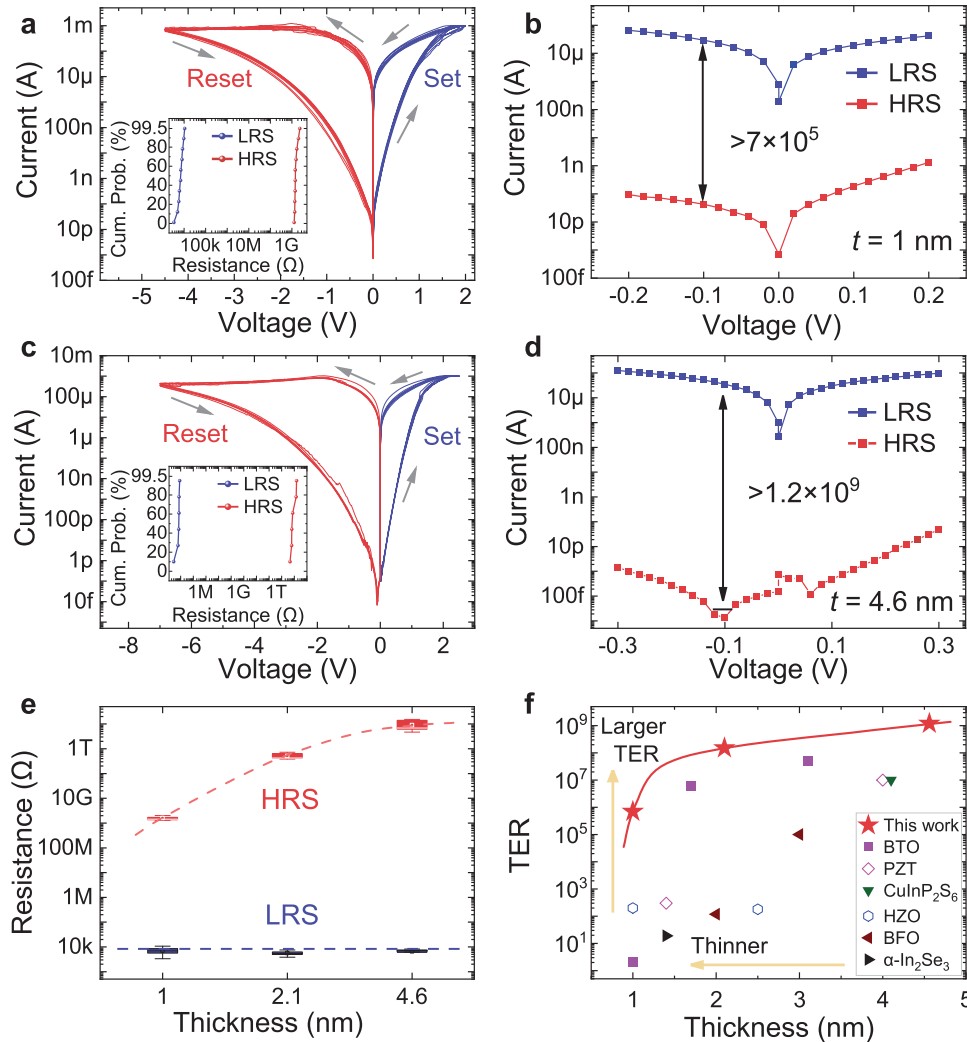

**Fig. 2 | Giant TER values in BSO-based FTJs. a** Quasi-static *I–V* sweeps for multiple cycles of a FTJ with BSO thickness of 1 nm. The voltages are applied on the metal side with the NSTO substrate grounded. Inset: cumulative probability distribution of LRS and HRS. **b** Over $7 \times 10^5$ TER achieved with −0.1 V read voltage. **c** Quasi-static *I–V* sweeps for multiple cycles of a FTJ with BSO thickness of 4.6 nm. Inset: cumulative probability distribution of LRS and HRS. **d** Over $1.2 \times 10^9$ TER achieved with −0.1 V read voltage. **e** Thickness effect on the TER values of BSO-based FTJs. The data of the FTJ with 2.1-nm BSO is extracted from Supplementary Fig. 5. The colored ranges in the box chart (25–75%) show the interquartile range (IQR), which is between the first and third quartile of the data distribution, and the error bars show the range within 1.5 IQR. **f** Comparison of TER values among ultra-thin FTJs with different ferroelectric materials and thicknesses. The purple squares are results of BTO-based devices from refs. 6,20,27. The magenta diamonds are results of PZT-based devices from refs. 29,30. The green triangle is result of the CuInP$_2$S$_6$ device from ref. 18. The blue hexagons are results of HZO-based devices from refs. 23,32. The brown triangles are results of BFO-based devices from refs. 28,31, and the black triangle is the result of the $\alpha$-In$_2$Se$_3$-based device from ref. 33. For FTJs with 1-nm-thick ferroelectric layer, our BSO-based FTJs show three orders of magnitude improvement in TER. Our FTJs with 4.6-nm-thick BSO show the highest TER of $1.2 \times 10^9$.

In the positive-up-negative-down (PUND) measurement mode (Supplementary Fig. 2), the FTJ with 1-nm-thick BSO layer shows a standard macroscopic hysteresis loop, by integrating the switching current curve (Fig. 1d). The remanent polarization up to 16.6 µC/cm$^2$ has been obtained, which is the foundation for achieving nonvolatile switching with a high TER. Piezoresponse force microscopy (PFM) measurements also demonstrate clear ferroelectric properties of the 1-nm-thick BSO films, with the out-of-plane phase showing obvious ferroelectric domain walls (Fig. 1e). The phase of the unbiased region is consistent with that of the region biased at +7 V, which proves that the ultrathin BSO films have intrinsic ferroelectric properties. The amplitude *vs.* voltage curve of the off-field local PFM presents a butterfly shape (Fig. 1f), which is a typical piezoelectric characteristic[24]. For BSO films with different thicknesses, the coercive fields and remanent polarizations are summarized in Supplementary Table 1.

## Giant TER of BSO-based FTJs

We leverage the high polarization of the BSO layer maintained at 1 nm thickness to achieve a high TER in FTJs. We employ the metal/ferroelectric/semiconductor device structure, which allows the modulation of both the height and width of the barrier[19]. The semiconductor substrate is STO with 0.7 wt% Nb doping, which provides low resistivity for decreasing the low-resistance state (LRS) resistance, and allows the efficient modulation of surface potential and the formation of the depletion region using the applied voltage or the polarization in the ferroelectric layer, as reported previously[20]. We first perform measurements using quasi-static voltage sweeps, by applying voltages on the metal side with the NSTO substrate grounded. These BSO-based FTJs show repeatable nonvolatile switching between the low-resistance state and high-resistance state (HRS) for multiple cycles (Fig. 2 and Supplementary Fig. 3). In FTJs with 1-nm BSO layer, we obtain a TER up to $7 \times 10^5$ by using 2 V set voltage and −4.5 V reset voltage (Fig. 2a, b).

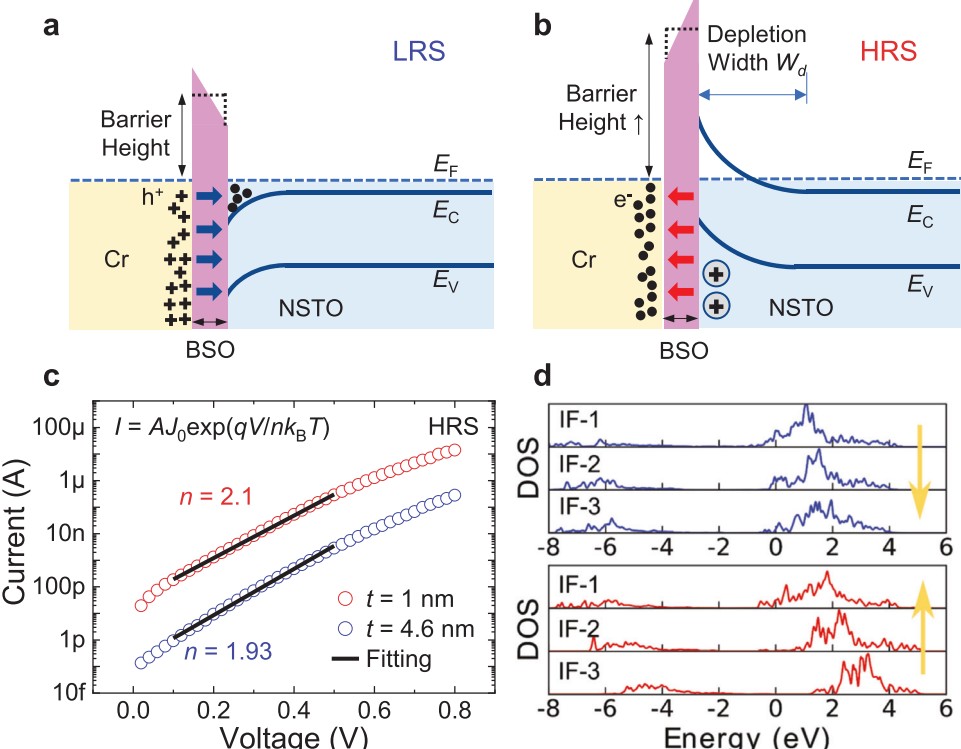

**Fig. 3 | Origin of giant TER. a, b** The energy band diagrams of FTJs in (**a**) LRS and (**b**) HRS states. In (**a**), the polarization is pointing to the degenerately doped NSTO side, and the positive bond charges in BSO lead to electron accumulation at the surface of NSTO. In this case, the electrons only need to tunnel through the ultra-thin ferroelectric barrier, and the FTJ is in LRS. In (**b**), the polarization is pointing to the metal side, and the negative bond charges in BSO will deplete the electrons at the surface of NSTO. In this case, an additional energy barrier at the F/S interface increases the effective tunnel barrier width, and thus the device is in HRS. **c** Fitting to the measured I−V curves for FTJs in HRS with 1-nm and 4.6-nm BSO layers, from

which the ideality factors are extracted as $n = 2.1$ and $n = 1.93$ for FTJs with 1-nm-thick and 4.6-nm-thick BSO, respectively, suggesting the existence of an extra energy barrier and thermally assisted tunneling mechanism in HRS. **d** The projected density of states onto the Ti 3d orbitals for the interface layers from DFT+U calculation. When the polarization is pointing to the NSTO side (downward arrow), all the interface (IF) layers are metallic. When the polarization is pointing to the Cr/Au side (upward arrow), some interface layers of NSTO convert into insulating due to the carrier depletion. IF-1 denotes the most neighboring NSTO layer adjacent to the ferroelectric layer.

The resistance shows gradual change with set and reset voltages without abrupt switching, and the switching characteristics are highly repeatable (inset of Fig. 2a). We also find that a higher reset voltage than set voltage is needed, because of the extra depletion region formed at the BSO/NSTO interface during the reset process, which will divide a portion of the total voltage, as reported previously[13,19]. Furthermore, for FTJs with 4.6-nm-thick BSO, the TER is $10^8$ with −6 V reset volage (Supplementary Fig. 4), and reaches up to $1.2 \times 10^9$ with −7 V reset voltage (Fig. 2c, d). The compliance current of 1 mA is applied in the set process to prevent the large tunneling current from affecting the device performance. The thickness effects on the HRS and LRS values are summarized in Fig. 2e, with the data of the FTJ with 2.1-nm BSO extracted from Supplementary Fig. 5. We observe that there is no significant change in the LRS (around 10 kΩ), while the HRS resistances increase with BSO thickness (from about 1.3 GΩ to 3.1 TΩ), which results in a higher TER with thicker BSO layers.

We benchmark the room-temperature TER values of our FTJs against those based on other types of ultrathin ferroelectric layers, including ABO-type perovskites (where A is a rare-earth or alkaline-earth metal, B is a transition metal, and O is oxygen)[6,20,27–31], binary oxides[23,32], and van der Waals layered materials[18,33] (Fig. 2f). Intriguingly, the high TER ($7 \times 10^5$) of the FTJs based on 1-nm BSO film is more than three orders of magnitude higher than previously reported TERs of FTJs based on other 1-nm-thick ferroelectric materials: the TER is only ~2 for FTJs based on 1-nm BTO[6], and is ~200 for FTJs based on 1-nm HZO[23]. While the metal/ferroelectric layer/semiconductor structure has been previously demonstrated[19,31,34], such high TER has not been reported for FTJs based

on 1-nm-thick ferroelectric films. When the thickness of the ferroelectric layer is down to 1 nm, the "dead layer" in the film, which has low dielectric constant can decrease the polarization[21]. In our device, Sm can help overcome the ferroelectric structural instability and depolarization field[24], and maintain a high polarization of 16.6 μC/cm² in BSO at 1 nm thickness. The high polarization of BSO is beneficial for modulating the barrier height and width, leading to such high TER values. We further demonstrate the highest TER up to $1.2 \times 10^9$ for FTJs based on the 4.6-nm-thick BSO film, which is more than 20 times higher than the state-of-the-art BTO-based FTJs (~5 × 10⁷)[27] and CuInP₂S₆-based FTJs (~10⁷)[18]. Such high TER values will decrease the leakage current and increase the reliability of the FTJs.

**Analysis of giant TER for BSO-based FTJs**

Analysis of the FTJ switching mechanism is demonstrated in Fig. 3. Upon contact with zero biasing, the different work functions between Cr (~4.5 eV) and the NSTO substrate will result in charge transfer, leading to a depletion region in NSTO and a built-in potential ($V_{bi}$) at the surface[19]. The energy band bending and energy barrier also changes with the polarization of the ferroelectric layer[20], and the effect is further enhanced with a thin BSO layer. When the BSO polarization is pointing from Cr/Au to NSTO, the positive bond charges in BSO will drive the electrons in NSTO to accumulate at the surface. The large polarization of BSO can eliminate the surface depletion in NSTO, and reduce the energy barrier (Fig. 3a). In this case, the FTJ is in LRS, and the structure is similar to metal/ferroelectric/metal FTJs, which supports large tunneling currents, with the I−V characteristics well fitted by the

direct tunneling (DT) model (Supplementary Fig. 6)[35]. When the polarization is pointing from NSTO to Cr/Au, the negative bond charges in BSO will enhance the depletion near the surface of the NSTO. In this case, the average ferroelectric barrier height increases. Meanwhile, an additional energy barrier at the ferroelectric/semiconductor (F/S) interface due to the surface depletion becomes more prominent in the conduction especially when the ferroelectric film is ultra-thin (Fig. 3b)[20]. The device is then in the HRS state, with the $I–V$ characteristics showing strong asymmetric and rectifying characteristics due to the F/S energy barrier (Fig. 2b, d). The ideality factors ($n$) are extracted as 2.1 and 1.93 for FTJs with 1-nm-thick and 4.6-nm-thick BSO, respectively (Fig. 3c), based on the junction transport model through an energy barrier (Supplementary Note 1). The deviation of $n$ from unity (ideal thermionic emission over a Schottky barrier) can be attributed to the thermally assisted tunneling in HRS, which has been observed in other NSTO-based Schottky-like junctions[36]. The large polarization of the BSO film effectively enhances both the average ferroelectric barrier height and the depletion region width, which highly suppress the HRS current, leading to larger TER values.

To confirm the electronic structure at the interface, we perform density functional theory plus Hubbard $U$ (DFT + $U$) calculations for the heterostructure model composed of one-layer bismuth oxide and four-layer NSTO (see Methods section). The interface model can be found in Supplementary Fig. 7. Because the Ti 3d states are the dominating states near the Fermi level in doped SrTiO$_3$[37,38], we show the projected density of states (DOS) onto the Ti 3d orbitals for the interface layers in Fig. 3d. In the case where the ferroelectric polarization is pointing toward the NSTO, all the interface layers are metallic. However, as the ferroelectric polarization is switched, some layers at the interface convert into insulating due to the carrier depletion, which implies that the width of the tunneling barrier will be increased according to previous reports[39]. Our DFT + $U$ calculations also confirm the elimination of energy barrier in LRS and the existence of extra barrier in HRS, consistent with the energy band diagrams shown in Fig. 3a, b. We further demonstrate that the effect of Sm atoms on the interfacial DOS and electronic properties at the interface is negligible (Supplementary Fig. 8). In addition, we have simulated the interface between bismuth oxide and Cr. The charge density difference in Supplementary Fig. 9 shows that the charge transfer can be modulated by the ferroelectric switching, which indicates that the Cr/BSO interface could also contribute to the large TER.

## Multi-level cells and analog memory properties of BSO-based FTJs

The giant TER combined with low variation can lead to multi-level cells towards high-density non-volatile memories on chip. By applying alternating reset and set pulses and measuring the resistances after the pulses, we demonstrate 20 cycles of switching in the FTJs with 1-nm-thick BSO ferroelectric layer (Fig. 4a). In addition, by fixing the set pulses at +2 V, and gradually increasing the reset voltage pulse amplitudes from −2 V to −4.5 V, the LRS of ~ 10 kΩ, and the HRS from 200 kΩ to 2 GΩ have been demonstrated. Then by fine-tuning the reset voltage from −1.5 V to −4.5 V with a step of −0.1 V, 32 distinct resistance states (5 bits of data storage) are achieved without any write-verification or write-termination processes (Fig. 4b). For each reset voltage, multiple cycles of resistive switching have been performed, and cycle-to-cycle variation has been summarized. While some resistance overlap exists between certain neighboring resistance states when considering the worst-case cycle-to-cycle variation, such small variation is tolerable for many in-memory computing applications[40]. When even higher accuracy of multi-level storage is necessary, reducing the resistance levels to 16 (4 bits of memory) can avoid resistance overlap even for the worst-case cycle-to-cycle variation. Other types of non-volatile memories such as resistive random-access memories can also realize similar resistance levels, but the closed-loop programming technique is used due to device variation

and nonlinear conductance tuning[41]. When even more resistance levels are desirable in our FTJs, write-verify or write-termination techniques can also be used to further reduce the variation, although there could be additional costs in circuit area, power, and delay[42,43].

We further explore linear tuning of multiple resistance states in the FTJs by gradually switching the ferroelectric domains[9,40]. We first measure the conductance states of BSO-based FTJs with increasing numbers of identical pulses, and find that the conductance is continuously modulated with +1.4 V pulses for potentiation and −1.6 V pulses for depression (Fig. 4c). The nonlinearity factor $\alpha$ is then extracted as 0.234 for potentiation and −0.494 for depression, where fully linear conductance modulation should have $\alpha$ of 0 (Supplementary Note 3). To further improve the linearity, we then apply another programming scheme: incremental step pulse programming (ISPP). The potentiation process uses a series of pulses with voltages increasing from +1.2 V to +1.95 V, with a step of +0.05 V, and the depression process uses pulses from −1.4 V to −2.9 V, with a step of −0.1 V (Fig. 4d). We observe repeatable and linear conductance modulation, with $\alpha$ extracted to be −0.106 for potentiation, and −0.195 for depression. The linear potentiation and depression are demonstrated in multiple FTJs using the ISPP programming scheme (Supplementary Fig. 10). The consistent linear conductance modulation in these FTJs is important not only for high-density memory, but also for in-memory computing and neuromorphic computing[9,29]. While the ISPP programming technique is more suitable for linear conductance modulation, the identical pulse technique can save the additional control circuits and energy consumption. Therefore, it could be preferred for resource-constrained systems.

The resistance states are also modulated by the voltage pulse duration. For a typical 1-nm-BSO-based FTJ, we find that the TER increases with a longer pulse duration (Fig. 4e). With +2 V set voltage and −4.5 V reset voltage, the device achieves TER over $10^5$ using 50 μs pulses, similar to the programming time for previously reported FTJs with similar device sizes[18,31,44]. We show that the TER is inversely proportional to the switching speed, and thus the operating speed can be highly improved if the device operates with a relatively small TER. As such, our device provides high flexibility of operation by trading off the TER and switching speed. It is also possible to use metal electrodes with lower work functions, or use smaller electrodes to further enhance the switching speed[40,45]. Using the ISPP programming technique, the resistance tuning range can be significantly enlarged by increasing the peak pulse amplitude ($V_P$) due to the stronger ferroelectric polarization (Fig. 4f and Supplementary Fig. 11).

## Endurance, retention, and device-to-device variation of BSO-based FTJs

Switching endurance, data retention, and device-to-device variation are critical for FTJs towards frequent set/reset, long-term stability, and large-scale integration[40]. We measure the endurance properties by first applying high programming voltage pulses of 2.5 V for set and −5 V for reset, and show TER over $10^5$ for about $5 \times 10^4$ endurance cycles (Supplementary Fig. 12). To achieve even higher endurance, we slightly decrease the programming voltages to 2.2 V for set and −3.5 V for reset, and demonstrate endurance up to $10^9$ cycles with no significant degradation of TER (TER over 300). In addition, the total switching endurance up to $5 \times 10^9$ with TER ~ 100 for the 1-nm-BSO-based FTJ is demonstrated (Fig. 5a). While the TER degrades after $5 \times 10^9$ cycles, the $I–V$ sweeps measured during the endurance switching cycles prove that the device is still functional (Supplementary Fig. 13), and the resistance window can be recovered by slightly increasing the voltage amplitude (Fig. 5b). The degraded TER could be attributed to the "dead layer" formed at the interface due to oxygen vacancy aggregation during the repetitively switching, which will limit the domain switching[34]. Furthermore, we measure the endurance of another FTJ based on 1-nm BSO, using 2.5 V set pulses and −5 V reset pulses until

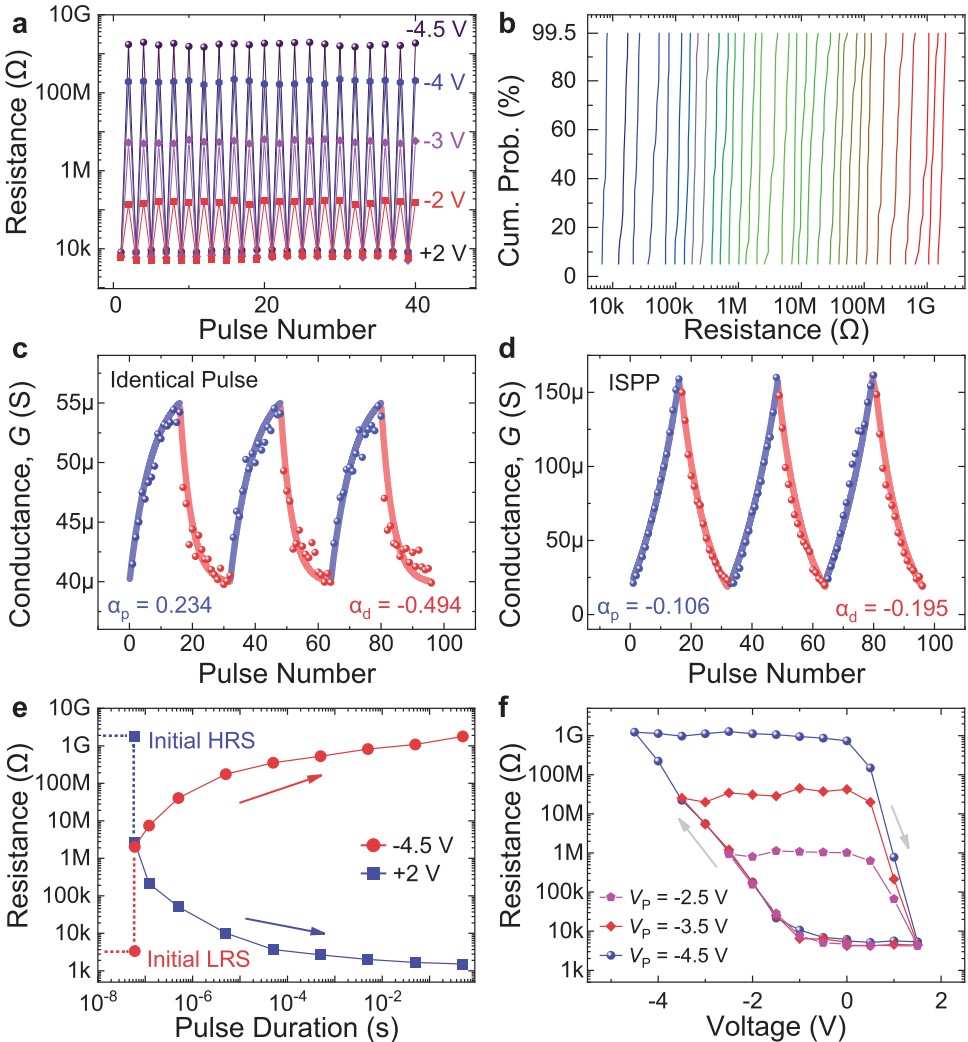

**Fig. 4 | Multi-level cell and analog properties of FTJs based on 1-nm-thick BSO.**
**a** Different resistance states achieved by applying reset pulses with different amplitudes. By applying alternating reset and set pulses and measuring the resistances after the pulses, we demonstrate repeatable resistive switching between LRS and different HRS states. **b** Cumulative probability distribution of 32 distinct resistance states achieved by applying increasing reset pulse amplitudes from −1.5 V to −4.5 V with a step of −0.1 V, obtained from multiple cycles of measurements for each reset pulse amplitude. **c** The conductance change with increasing pulse numbers, using identical pulses of +1.4 V for potentiation and −1.6 V for depression.

The pulse duration is $t_d$ = 60 ns. **d** Improved potentiation and depression process using ISPP, showing repeatable and linear conductance modulation. The potentiation process uses a series of pulses with voltages increasing from +1.2 V to +1.95 V, with a step of +0.05 V, and the depression process uses pulses from −1.4 V to −2.9 V, with a step of −0.1 V. **e** Resistance change with pulse duration using +2 V set pulses and −4.5 V reset pulses, showing that the TER increases with the pulse duration, and TER over $10^5$ can be achieved using 50 μs pulses. **f** Resistance change with voltage measured using the ISPP scheme, showing a larger resistance tuning range when gradually increasing the peak pulse amplitude ($V_P$).

the device degrades (Supplementary Fig. 14). We find that the TER remains above $2 \times 10^5$ for over $10^5$ cycles, and then starts to degrade. Nevertheless, the total endurance cycle over $5 \times 10^9$ cycles is still demonstrated. The higher endurance of BSO-based FTJs compared with that of previous reports (Supplementary Table 2) could be related to the high-quality and stable BSO film with less grain boundaries[24,46].

The retention properties are measured for more than 10 hours for both LRS and HRS (Fig. 5c), which translates to over $10^5$ TER maintained for more than 10 years. We also measure the retention properties for several intermediate states (Supplementary Fig. 15), showing consistent and stable resistance states. The long retention of these BSO-based FTJs could be attributed to the Sm-induced stable tetragonal-like phase structure[24], which minimizes the degradation of polarization due to the depolarization field as in previous ultra-thin FTJs[46,47]. We further measure the HRS, LRS, and TER of 35 FTJs with 1-nm BSO layer (Fig. 5d). High uniformity among different devices with TER over $10^5$ has been demonstrated, which is important towards large-scale array integration[10].

## Discussion

The NSTO substrate has been a common substrate for constructing high-performance FTJs[6,19,45]. While the depletion region at the surface of NSTO may lead to a larger resistance compared with a metal contact, the high switching speed of 300 ps and low programming energy below 10 fJ have been demonstrated in FTJs using the NSTO substrate[40]. Furthermore, although currently the NSTO substrate instead of Si substrate is used, it is promising to obtain BSO films on substrates compatible with complementary−metal−oxide−semiconductor (CMOS) fabrication processes. For example, ultrathin $SrTiO_3$ or similar films can be grown on silicon-on-insulator (SOI) wafers using molecular beam epitaxy (MBE)[48,49], or bonded to the Si substrate through the ion slicing, or the "smart-cut" process[50,51], which could also be used for BSO film growth. Furthermore, by using a water-soluble $Sr_3Al_2O_6$ as the sacrificial layer, the $SrTiO_3$ and $BiFeO_3$ thin films have been released from the growth substrate in water, and transferred to a Si substrate. Such a technique may also be used to transfer the BSO films. Besides, we demonstrate the initial results for the growth of the BSO film on metal-on-oxidized-Si

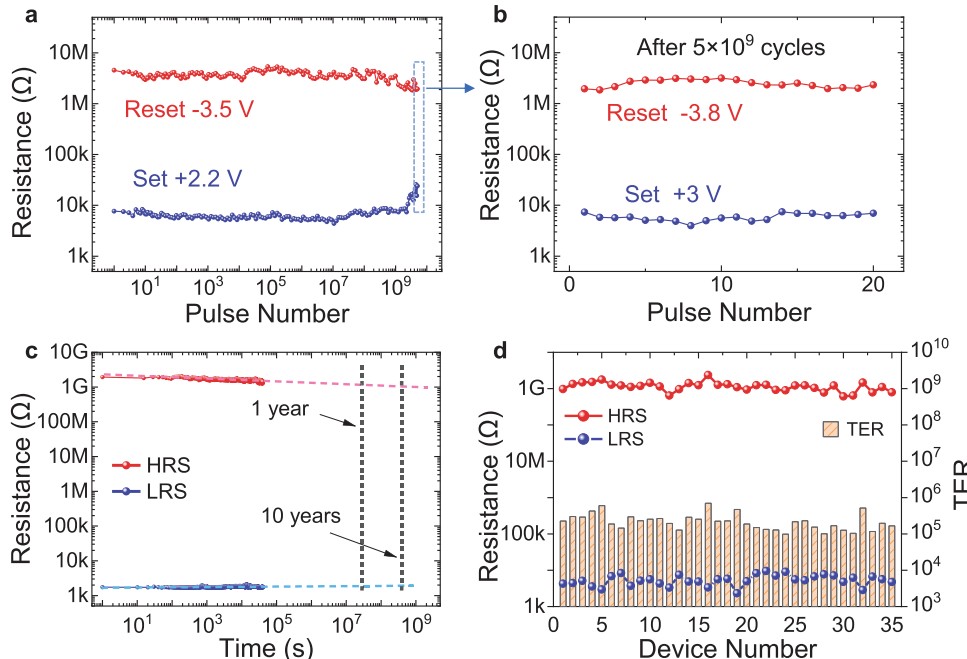

**Fig. 5 | Endurance, retention, and device-to-device variation of FTJs based on 1-nm-thick BSO. a** Measured endurance properties showing more than $5 \times 10^9$ cycles under +2.2 V/−3.5 V programming pulses. **b** HRS and LRS resistances after $5 \times 10^9$ programming cycles, showing that the device can maintain TER over 100 by applying slightly larger programming voltages. **c** Measured retention property showing that TER over $10^5$ can be maintained for more than 10 years. **d** HRS, LRS, and TER measured for 35 devices, showing small device-to-device variation.

substrates such as the Au/SiO$_2$/Si substrate (Supplementary Fig. 16). These results show high potential for the integration of BSO films onto Si substrates, and for on-chip memories using BSO-based FTJs.

In oxide-based materials such as BSO, the filamentary switching due to vacancy motion should also be considered. However, in these BSO-based FTJs, we do not observe the initial forming process that is required for most of the filamentary switching memories without special forming-free designs[52,53]. We also measure the temperature-dependent $I$–$V$ characteristics in both LRS and HRS (Supplementary Fig. 17), which shows that the current increases with temperature in both HRS and LRS, with a larger current increase in HRS. Such trend in LRS is contrary to the filamentary switching which should have a decreasing current with higher temperature due to the metallic conduction through the conductive filament[54]. Through fitting, we find that direct tunneling is dominant in LRS, and thermally assisted tunneling is dominant in HRS, with the energy barrier of 0.1 eV extracted for LRS, and 1.25 eV extracted for HRS, consistent with ferroelectric switching (Supplementary Note 1.2). Furthermore, we measure the capacitances of the FTJs in LRS and HRS (Supplementary Fig. 18). The lower capacitance in HRS than that in LRS, and the capacitance–voltage relationship that fits well to the depletion capacitance model suggest the existence of a built-in potential and a depletion region (Supplementary Note 2). Therefore, ferroelectric switching should be the dominant mechanism in these BSO-based FTJs.

In summary, we demonstrate that BSO-based FTJs with down to one-nanometer ferroelectric layer can maintain a large TER over $7 \times 10^5$, which is at least three orders of magnitude higher than the TER of other FTJs with 1-nm-thick ferroelectric barrier. The FTJs with 2.1-nm and 4.6-nm BSO layers achieve TER values over $10^8$ and $10^9$, respectively, which are higher than commercial flash memories. The DFT + $U$ calculations confirm that the effective modulation of energy barrier owing to the large polarization of the BSO layer is essential to achieve the high TER. These BSO-based FTJs also show desirable analog memory properties by exhibiting 32 resistance levels and high linearity in conductance tuning. Furthermore, they show high endurance cycle,

long retention, and small device-to-device variation, making them highly promising for low-power, high-reliability, and high-density non-volatile memories and in-memory computing.

## Methods
### Material growth and device fabrication
BSO films were prepared by chemical solution method. Bi(NO$_3$)$_3$·5H$_2$O and Sm(NO$_3$)$_3$·6H$_2$O were dissolved in glycol methyl ether solvent at the molar ratio of 10:1 in turn, thoroughly mixed and left for 4 h. The global bottom electrode is the conductive substate NSTO (0.7 wt% Nb-doped SiTiO$_3$) with a thickness of 500 μm. The precursor solution was dropped onto the substrate and spun for 30 s at the speed of 5000 rpm. Then annealing was performed at 90, 270, and 600 °C, with the annealing time of 10 min, 5 min, and 30 min, respectively. On the epitaxially grown BSO film, the top circular 15 nm Cr/60 nm Au electrodes with diameters of 10 μm were lithographically defined.

### Material characterization
The TEM images of the BSO-based FTJs were obtained by scanning the specimens using the Titan G2 60–300 instrument, where the specimens were prepared by a focused ion beam tool (FEI Helios NanoLab 600i) operated at the voltage of 2–30 kV. The polarization−electric field ($P$ − $E$) loops were measured using the ferroelectric measurement system (aixACCT TF-Analyzer 3000) at the frequency of 0.01 − 1 MHz. Atomic force microscopy machines from both Asylum Research (MFP3D) and Bruker (Dimension Icon) were used to obtain the piezoresponse force microscopy images. The box-in-box patterns were written on the BSO film by applying the voltage of +7 V or −7 V to the corresponding regions, respectively. The X-ray diffraction (XRD) pattern of the BSO film was collected by an X-ray diffractometer (PW 3040-X'Pert Pro, PANalytical, Cu Kα, $\lambda = 1.54$ Å).

### Electrical characterization
Electrical measurements of the FTJs were performed using a Keithley 4200A-SCS semiconductor parameter analyzer (for $I$ − $V$ measurements)

and Agilent B1500A semiconductor parameter analyzer (for $C - V$ measurements) connected to a probe station under atmospheric pressure and at room temperature. For the BSO-based FTJs, source measurement units (SMUs) with preamplifiers were used in quasi-static electrical measurements, pulse measurement units (PMUs) with remote pre-amplifier (RPM)/switch modules were used for the pulsed measurements, and the multi-frequency capacitance measurement units (MFCMUs) were used for $C - V$ measurements. During the measurements, the two probes were connected to the top electrode on BSO film and the NSTO substrate under BSO layer, respectively.

## First-principles calculations

The VASP code[55–57], incorporated with Perdew-Burke-Ernzerhof (PBE) functionals and a cutoff of 450 eV, was employed to carry out the non-spin polarized density functional theory calculations. To study the electronic structure at the interface between the ferroelectric layer and NSTO, we employed a supercell structure composed one-unit cell of ferroelectric bismuth oxide and four layers of NSTO. To simulate the epitaxial growth of the bismuth oxide on NSTO, the in-plane lattice constant $a$ was constrained to the optimized lattice constant of NSTO (3.978 angstrom) from our first-principles calculations. We used the rotationally invariant DFT + $U$ method[58] to take into account correlation effects on 3d orbitals of transition metal ions. The commonly used $U = 5.0$ eV and $J = 0.64$ eV were applied to Ti and Nb 3d orbitals[59]. In the self-consistent field and DOS calculations, the k-grids of $8 \times 4 \times 1$ and $12 \times 6 \times 2$ were used, respectively. Since the concentration of Nb in the actual NSTO was very low (0.7% in weight percent) and cannot be well simulated by standard first-principles calculations, we replaced one Ti atom out of eight by one Nb atom to model the Nb doped $SrTiO_3$. Specifically, one Ti atom at the interface was replaced with one Nb atom.

## Data availability

All data needed to evaluate the conclusions in the paper are present in the paper and/or the Supplementary Information. All other relevant data of this study are available from the corresponding author upon reasonable request.

## Code availability

All relevant code or algorithm are available from the corresponding author upon reasonable request.

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

## Acknowledgements
The work was supported by the National Natural Science Foundation of China (NSFC) (Grants 62250073, 62104140, U21A20505, 92364107, 22371013), the Fundamental Research Funds for the Central Universities (FRF-IDRY-19-007 and FRF-TP-19-055A2Z), Science and Technology Commission of Shanghai Municipality (STCSM) Shanghai Rising-Star Program (Grant 23QA1405300) and Natural Science Project General Program (Grant 21ZR1433800), Lingang Laboratory Open Research Fund (Grant LG-QS-202202-11), National Program for Support of Top-notch Young Professionals, the Young Elite Scientists Sponsorship Program by CAST (2019-2021QNRC), the "Xiaomi Young Scholar" Funding Project, and Natural Science Foundation of Chongqing (CSTB2022NSCQ-MSX1095). Use of the Beijing Synchrotron Radiation Facility (1W1A beamlines, China) of the Chinese Academy of Sciences is acknowledged. R.Y. and Y.J. thank the Center for Advanced Electronic Materials and Devices (AEMD) of Shanghai Jiao Tong University, for the support in the fabrication and measurements of the devices.

## Author contributions
L.Z. and R.Y. conceived the whole research and supervised the whole research activities. Y.J. performed the device fabrication and measurements. Q.Y. performed the material growth and characterization. Y.-W.F. performed the theoretical calculations. Y.L. performed the TEM characterization. M.X. and J.T. assisted in the data analysis. J.W. assisted in the temperature-dependent measurements. Y.J., Q.Y., Y.-W.F., L.Z. and R.Y. wrote the manuscript with input and comments from all the authors.

## Competing interests
The authors declare no competing interests.
