## [Peer Review File · Nature Communications]

REVIEWER COMMENTS

Reviewer #1 (Remarks to the Author):

The authors reported the demonstration of FTJ device based on metal/BSO/NSTO with high TER ratio. The reviewer has concerns about quality of the data presented in the manuscript as well as the novelty of this work as compared to previous reports in the literature. First, the ferroelectricity in the BSO material has already been reported in the authors' previous publication (Yang et al., Science 379, 1218–1224 (2023)). Fig.1 of this manuscript is just a repetition of their previous work. Secondly, there has already been previous reports on using metal/BaTiO₃/NSTO FTJ device structure to achieve high TER (Nat. Commun. 8, 15217 (2017)), and there is no innovation in the device structure except for replacing the BaTiO₃ with BSO.

Moreover, in general, the use of NSTO semiconducting electrode to boost the TER in FTJ is not preferable for practical applications due to the larger resistance introduced by the semiconducting contact as compared to metal contact, which can limit its operation speed and increase programming/read energy loss. The device structure is also not compatible with the CMOS process for practical memory applications as the BSO has to be grown on the NSTO perovskite substrate.

The reviewer also has the following technical concerns:

- The authors need to show evidence that the memory switching characteristics in the device is not due to other type of transport mechanism such as by filamentary switching. Filamentary switching can be possible in oxide materials like the BSO. Such additional measurement that should be carried out include temperature dependent I-V characteristics.

- In Fig. 1, it seems that the coercive field is less than 2 V for the 1 nm BSO material. Why the reset voltage in the device is much higher (-4.5 V) for the FTJ device. The set and reset voltages are also significantly asymmetrical. The authors should clarify this.

- Fig. 5, the device used for endurance test in Fig 5a and 5b only has TER less than 1000. The author should use device with the claimed high TER for such tests. Also, the device used for the endurance test (Fig. 5a and 5b) and the device used for the retention test (Fig. 5c) has very different TER ratio and the data apparently comes from two different devices. It is important that for the 1 nm BSO thickness FTJ device, the I-V characteristics in Fig. 2a and 2b, the multi-level test in Fig. 4, the endurance and the retention test should be done on the same device. A good FTJ needs simultaneously have good features in these tests.

- The coercive field and remnant polarization of the BSO material used in this work needs to be stated in the manuscript.

- Fig. 1c, the TEM image seems to show a transition region at the interface of BSO and Au/Cr metal (or the BSO layer seems to be thicker than 1 nm). The authors need to clarify this.

Reviewer #2 (Remarks to the Author):

In the present manuscript, the authors designed FTJs based on Sm-substituted layered bismuth oxide (BSO). They demonstrated that these FEJs can achieve a TER over 7×10^5 with a 1-nm ferroelectric BSO film and three orders of magnitude higher than previous reports with such thickness. They also showed that the TER can be further enhanced as the thickness of the BSO layer increases and these FTJs demonstrate up to 32 resistance states. The experimental data are interesting and exciting. However, I

think the theoretical explanation is not robust enough to support the present experimental findings. Therefore, I do not think the current version of the manuscript is suitable for publication in Nature Communications. Below are the reasons.

1. In the DFT calculation, the authors employed a supercell structure composed one-unit cell of ferroelectric bismuth oxide and four layers of NSTO to study the electronic structure of the interface between the ferroelectric layer and NSTO. Please provide a detailed interface atomic structure model.
2. The authors used the DFT+U method, which did not consider the U value of Sm atom, but only considered Ti atom. The existence of f electrons in Sm atoms is difficult to converge, and the U value needs to be considered.
3. How do theoretical calculations simulate the flipping of the ferroelectric polarization direction of BSO? Is it just to rotate the BSO 180 degrees? Does it match the experiment? Please give a detailed explanation.
4. The authors only provided the electronic properties of the BSO-NSTO interface in flipping the ferroelectric polarization direction, how about the BSO-Cr/Au interface? Perhaps differential charge densities would provide more robust evidence. Please give the corresponding result.
5. In Figure 3c and Figure S5, the I-V curve fitted by the direct tunneling model is extremely perfect with the experimental results, which makes one have to doubt the authenticity of the theoretical fitting data. Can a more accurate theoretical calculation be given based on the non-equilibrium Green's function method?
6. The colors of Sm atoms and Bi atoms in the atomic structure model in Fig.1 (b) are difficult to distinguish, please modify.
7. Tunneling or Tunnelling, Programming or Programing? Please use consistent spelling throughout the manuscript.

Reviewer #3 (Remarks to the Author):

Jia et al. report on the novel BSO-based ferroelectric tunnel junctions that can maintain a large TER over 10^5 even with the film thickness down to 1 nm, three orders of magnitude higher than previous results at similar film thickness. Previously, the scaling down of FTJ is highly limited by the critical-thickness effect, because the polarization is shielded as the thickness decreases due to the huge depolarization field. Therefore, the demonstration of such large TER ratio at such small thickness makes a significant advance to the field, and is meaningful to the community. The reliable multi-level cells and analog properties also make these devices promising for various applications. The manuscript is well organized, with solid experimental results and theoretical analysis. After the following issues are further explained and addressed, the manuscript should be suitable for publication in Nature Communications.

1. In Fig. 1b, it seems that the Bi/Sm atoms are not aligned, and the Sm atoms only appear in the upper half of the crystal structure, while there is no explanation about the structure. Besides, the thickness of the BSO layer shown in Fig. 1c seems to be slightly larger than 1 nm. Why it is not consistent with the X-ray reflectivity measurement results?
2. In the local PFM measurement shown in Fig. 1f, the phase change is $\sim 120^\circ$ instead of the typical 180° , and the saturation in phase is observed. Is there any explanation on this effect?
3. The authors use STO substrate with 0.7 wt% Nb doping. Why is such doping concentration used,

and have the authors considered using substrates with other doping concentrations?

4. In P.13, Line 230, the authors claim "32 distinct resistance states (5 bits of data storage) are achieved without any write-verification or write termination processes". While 32 MLC achieved by simply varying reset voltages is remarkable, in Fig. 4b there seems to be some overlap among the neighboring resistance levels with variation considered. Will this affect the accuracy for memory/computing application, and are there any solutions that can avoid this issue?

5. The authors claim that they explore the linear tuning of multiple conductance states using two techniques: identical pulses and ISPP. It seems obvious from Fig. 4c-d that the results obtained by ISPP method are much more linear than those obtained by the identical pulse method. Is the identical pulse method still useful for programming these devices then?

6. In Fig. 5a-b, the TER clearly degrades after 5×10^9 endurance cycles. What is the physical mechanism of the degraded TER? What are the factors that can affect the endurance cycle in these FTJs? These should be more clearly explained.

7. Some important references about ferroelectric switching are missing (e.g. "Intrinsic ferroelectric switching from first principles", Nature 2016; "Emergence of room-temperature ferroelectricity at reduced dimensions", Science 2015).

Response to Reviewers' Comments – NCOMMS-23-26395-T

Dear Reviewers:

This document presents our full response to the Reviewers' Comments – we have endeavored to address all the technical issues raised. We highly appreciate the feedback from all Reviewers, and the constructive comments and suggestions. In response to Reviewers' questions and comments, we have made great efforts in clarifying and explaining the technical issues raised, by providing additional solid evidence. The main changes include:

- (1) Addition of explanation about the novelty of this work;
- (2) Addition of Supplementary Fig. 16, relevant discussions, and references explaining the operating speed, energy, and potential for CMOS-compatibility of BSO-based FTJs;
- (3) Addition of temperature-dependent I - V measurements in Supplementary Fig. 17 and explanation for confirming the switching mechanism;
- (4) Addition of capacitance measurements at different frequencies for HRS and LRS in Supplementary Fig. 18 for confirming the depletion region change during ferroelectric switching;
- (5) Additional explanation about the higher reset voltage than the coercive field;
- (6) Addition of endurance measurements on another 1-nm-BSO-based FTJ with high TER in Supplementary Fig. 14;
- (7) Addition of the summary for coercive fields and remanent polarizations for BSO films with different thicknesses in Supplementary Table 1;
- (8) Addition of the BSO atomic model in Supplementary Fig. 1 and discussions on film thickness;
- (9) Addition of the interface atomic structure model in Supplementary Fig. 7;
- (10) Addition of DFT calculation for the BSO/NSTO interface in Supplementary Fig. 8;
- (11) Addition of text explaining the simulation of ferroelectric polarization flipping in theoretical calculation;
- (12) Addition of calculation for the charge difference at the interface between bismuth oxide and Cr at different directions of polarization in Supplementary Fig. 9;

- (13) Addition of explanation for the fitting of measured I – V curves and the reference;
- (14) Revision of the color for the atomic structure model in Fig. 1b;
- (15) Addition of explanations about the resistance variation and multi-level storage;
- (16) Addition of explanations about the different properties between ISPP and identical pulse programming techniques;
- (17) Addition of explanations about the degradation of TER during endurance measurements;
- (18) Addition of references about ferroelectric switching.

All changes are highlighted in the revised Main Text and Supporting Information.

The responses to every point from all Reviewers are detailed below.

(I) Response to Reviewer #1

Here we present our point-by-point response to Reviewer #1 below. The original comments are in green color and in Calibri font; responses are in black color and Times New Rome font.

Reviewer #1: "The authors reported the demonstration of FTJ device based on metal/BSO/NSTO with high TER ratio. The reviewer has concerns about quality of the data presented in the manuscript as well as the novelty of this work as compared to previous reports in the literature. First, the ferroelectricity in the BSO material has already been reported in the authors' previous publication (Yang et al., *Science* 379,1218–1224 (2023)). Fig.1 of this manuscript is just a repetition of their previous work. Secondly, there has already been previous reports on using metal/BaTiO₃/NSTO FTJ device structure to achieve high TER (*Nat. Commun.* 8, 15217(2017)), and there is no innovation in the device structure except for replacing the BaTiO₃ with BSO.

We thank Reviewer #1 for raising this important issue, which allows us to further clarify the novelty of our work. First of all, we would like to clarify that our work is very different from our previous publication. In Ref. 24 in the Main Text [Yang, Q. et al. *Science* **379**, 1218–1224 (2023)], we mainly focused on the discovery of the new ferroelectric material: layered bismuth oxide through samarium substitution [Bi_{1.8}Sm_{0.2}O₃ (BSO)], by performing the BSO material growth and material-oriented characterization to study the origin of ferroelectricity in BSO materials when the thickness scales down. While in this work, we focus on the ferroelectric tunnel junction (FTJ) devices based on BSO, with measurements on electrical properties towards memory and in-memory computing, and theoretical analysis using DFT+*U* first-principles calculations. By leveraging the large ferroelectric polarization maintained at 1 nm thickness in BSO films, we solve the problem of limited tunnelling electroresistance (TER) in nanometer-thickness FTJ devices, and demonstrate a giant TER up to 7×10^5 in these atomic-scale FTJs, which is three orders of magnitude higher than that in previous FTJs with 1-nm-thick ferroelectric layers. We further demonstrate low cycle-to-cycle and device-to-device variations, high linearity of conductance modulation or excellent analog memory properties, as well as high endurance cycles over 5×10^9 , which show the high promise of BSO-based FTJs for memory and in-memory computing applications, and have not been reported in previous works.

Second, while we agree with Reviewer #1 that the metal/ferroelectric layer/NSTO FTJ structure with BaTiO₃ switching layer has been reported as shown in [Ref. 20: Xi, Z. et al. *Nat. Commun.* **8**, 15217 (2017)], we respectfully disagree with Reviewer #1's assertion that "there is no innovation...". We strongly believe that our BSO-based FTJ is a significant advance and innovation in the field, and the BSO-based FTJs in this work outperform the BaTiO₃-based FTJs in various aspects, as summarized below.

- (1) At atomic-scale thickness (~1 nm), BaTiO₃-based FTJs show TER of only ~2, due to the ferroelectric structural instability and large depolarization field [Ref. 6: Garcia, V. et al. Giant tunnel electroresistance for non-destructive readout of ferroelectric states. *Nature* **460**, 81–84 (2009)]. In contrast, the BSO-based FTJs with 1 nm BSO thickness in this work show TER up to 7×10^5 , which benefits from the strong ferroelectric polarization and shows superior performance.
- (2) Macroscopic ferroelectric hysteresis loops have not been demonstrated in ultrathin BaTiO₃ films, to the best of our knowledge. In contrast, we successfully demonstrate the ferroelectric hysteresis loop for BSO films down to 1-nm thickness in Fig. 1d in the Main Text, which shows remanent polarization up to $16.6 \mu\text{C}/\text{cm}^2$, and can be used to directly identify the ferroelectricity.
- (3) When the film thickness gets thicker than 1 nm, while both BaTiO₃-based FTJs and our BSO-based FTJs show higher TER values than the 1-nm counterparts, the highest reported TER of BaTiO₃-based FTJ is $\sim 5 \times 10^7$, to the best of our knowledge. In contrast, the BSO-based FTJs in this work show higher TER values up to 1.2×10^9 , which is more than 20 times higher than the state-of-the-art BaTiO₃-based FTJs (Fig. R1).

Figure R1. Comparison between BSO-based FTJs and BaTiO₃-based FTJs.

- (4) The BaTiO₃-based FTJs with 2.4-nm-thick ferroelectric layer show the endurance cycle of 10⁸–10⁹ [Ref. 44: Ma, C. et al. Sub-nanosecond memristor based on ferroelectric tunnel junction. *Nat. Commun.* **11**, 1439 (2020)]. In contrast, the BSO-based FTJs with 1-nm-thick ferroelectric layer in this work show endurance cycle over 5×10⁹. In addition, we have summarized the endurance of ultra-thin FTJs based on BTO or other materials in Supplementary Table 2, which shows that the endurance of BSO-based FTJs in this work is indeed among the highest. This is advantageous for increasing the device lifetime.
- (5) Our BSO-based FTJs show up to 32 resistance states with high linearity of conductance modulation, without using any write-verify technique, making them highly promising for multi-level memories and in-memory computing.

Finally, we would like to clarify that the metal/ferroelectric layer/semiconductor structure is a very commonly used structure for FTJs with a high TER, and a number of interesting progresses have been reported based on such structure, such as [Ref. 19: Wen, Z. et al. Ferroelectric-field-effect-enhanced electroresistance in metal/ferroelectric/semiconductor tunnel junctions. *Nat. Mater.* **12**, 617–621 (2013); Ref. 31: Hu, W. J. et al. Optically controlled electroresistance and electrically controlled photovoltage in ferroelectric tunnel junctions. *Nat. Commun.* **7**, 10808 (2016); Ref. 45: Yang, Y. et al. Atomic-scale fatigue mechanism of ferroelectric tunnel junctions. *Sci. Adv.* **7**, eabh2716 (2021)]. Therefore, we believe that our work based on a similar device structure but with much improved FTJ device properties shows important and timely results for advancing the field.

In summary, we demonstrate the giant TER, high endurance, and desirable analog memory properties in BSO-based FTJs, using both abundant experimental characterizations and DFT+*U* calculations. We believe that our work will be a significant advance for the field as the first demonstration of BSO-based FTJ, and for solving the critical issue of low TER when the ferroelectric layer thickness is down to the atomic scale.

We have added the following text and reference in the revised manuscript to help clarify this point, which we reproduce below:

“While the metal/ferroelectric layer/semiconductor structure has been previously demonstrated^{19,31,45}, such high TER has not been reported for FTJs based on 1-nm-thick ferroelectric films.”

“We measure the endurance properties by first applying high programming voltage pulses of 2.5 V for set and -5 V for reset, and show TER over 10^5 for about 5×10^4 endurance cycles (Supplementary Fig. 12). To achieve even higher endurance, we slightly decrease the programming voltages to 2.2 V for set and -3.5 V for reset, and demonstrate endurance up to 10^9 cycles with no significant degradation of TER (TER over 300). In addition, the total switching endurance up to 5×10^9 with TER ~ 100 for the 1-nm-BSO-based FTJ is demonstrated (Fig. 5a). While the TER degrades after 5×10^9 cycles, the *I-V* sweeps measured during the endurance switching cycles prove that the device is still functional (Supplementary Fig. 13), and the resistance window can be recovered by slightly increasing the voltage amplitude (Fig. 5b).”

“44 Ma, C. et al. Sub-nanosecond memristor based on ferroelectric tunnel junction. *Nat. Commun.* **11**, 1439 (2020).”

Reviewer #1: Moreover, in general, the use of NSTO semiconducting electrode to boost the TER in FTJ is not preferable for practical applications due to the larger resistance introduced by the semiconducting contact as compared to metal contact, which can limit its operation speed and increase programming/read energy loss. The device structure is also not compatible with the CMOS process for practical memory applications as the BSO has to be grown on the NSTO perovskite substrate.”

We thank Reviewer #1 for raising this important issue, which allows us to further clarify the selection of semiconducting electrodes. First of all, as Reviewer #1 has mentioned in the previous

comment, the NSTO semiconducting electrode has been a common selection in various works [Ref. 6: Garcia, V. et al. Giant tunnel electroresistance for non-destructive readout of ferroelectric states. *Nature* **460**, 81–84 (2009); Ref. 19: Wen, Z. et al. Ferroelectric-field-effect-enhanced electroresistance in metal/ferroelectric/semiconductor tunnel junctions. *Nat. Mater.* **12**, 617–621 (2013); Ref. 44: Ma, C. et al. Sub-nanosecond memristor based on ferroelectric tunnel junction. *Nat. Commun.* **11**, 1439 (2020)]. These works provide significant advances to the community of ferroelectric tunnel junctions. Therefore, the problems raised by Reviewer #1 are common issues for the whole field, instead of our work in specific.

Second, while we agree with Reviewer #1 that the depletion region of the NSTO semiconducting electrode can lead to a larger resistance compared with the metal electrode, we believe that such semiconducting contact does not necessarily lead to a low speed and high energy consumption. For example, in the Ag/PbZr_{0.52}Ti_{0.48}O₃ (PZT)/NSTO FTJ, subnanosecond programming speed with the fastest switching speed of 300 ps has been demonstrated, and programming energy below 10 fJ has been achieved in the FTJ with 50-nm diameter [Ref. 42: Luo, Z. et al. High-precision and linear weight updates by subnanosecond pulses in ferroelectric tunnel junction for neuro-inspired computing. *Nat. Commun.* **13**, 699 (2022)]. Furthermore, we indeed find that there is a trade-off between the TER and operating speed. As shown in Fig. 4e in the Main Text, a longer pulse duration leads to a larger TER. Therefore, depending on the need for the specific application, we can either program the device with high speed when a high TER is not necessary, or achieve a higher TER with extra programming time. Such trade-off has been reported for other FTJs as well [Ref. 18: Wu, J. et al. High tunnelling electroresistance in a ferroelectric van der Waals heterojunction via giant barrier height modulation. *Nat. Electron.* **3**, 466–472 (2020)].

Third, while we agree with Reviewer #1 that currently the NSTO substrate instead of Si substrate is used, we would like to clarify that the following progresses have made the BSO films and FTJ devices highly promising for constructing CMOS-compatible FTJs.

- (1) It is possible to achieve thin-film crystalline NSTO or BSO layers on silicon-on-insulator (SOI) substrates. Ultrathin SrTiO₃ (STO) has been grown on SOI wafers using molecular beam epitaxy (MBE), and then the ultrathin crystalline Si layer is oxidized through a dry O₂ annealing process, resulting in crystalline STO layer on oxidized Si substrate, which

can then be used as the substrate for BSO growth [Ref. 48: Ortmann, J. E. et al. Epitaxial Oxides on Glass: A Platform for Integrated Oxide Devices, *ACS Appl. Nano Mater.* **2**, 7713–7718 (2019)]. Similarly, single-crystalline BaTiO₃ has also been epitaxially grown on SOI substrate [Ref. 49: Xiong, C. et al. Active Silicon Integrated Nanophotonics: Ferroelectric BaTiO₃ Devices, *Nano Lett.* **14**, 1419–1425 (2014)].

- (2) The thin film crystalline STO on Si substrate can also be achieved through the ion slicing, or the “smart-cut” process. The general procedure includes H⁺ ion implantation into a bulk single crystal STO wafer, bonding of the implanted STO to another substrate which could be glass on Si, an annealing process to enable the slicing, and a polishing process to thin down the film and make the surface smooth, as reported in [Ref. 50: Roth, R. M. et al. Fabrication and material properties of submicrometer SrTiO₃ films exfoliated using crystal ion slicing, *Appl. Phys. Lett.* **90**, 112913 (2007); and Ref. 51: Lee, Y. S. et al. Fabrication of patterned single-crystal SrTiO₃ thin films by ion slicing and anodic bonding, *Appl. Phys. Lett.* **89**, 122902 (2006)].
- (3) Furthermore, it is possible to transfer the grown BSO film to a CMOS-compatible substrate. By using a water-soluble Sr₃Al₂O₆ as the sacrificial buffer layer, the SrTiO₃ and BiFeO₃ thin films have been released in water, and transferred to a Si substrate using a PDMS stamp [Ref. 26: Ji, D. et al. Freestanding crystalline oxide perovskites down to the monolayer limit. *Nature* **570**, 87–90 (2019)].
- (4) While our work is focused on the first experimental demonstration of BSO-based FTJ with high TER and other desirable properties, we have obtained some preliminary results on the growth of BSO films on other metal films on oxidized Si substrates such as the Au/SiO₂/Si substrate, as shown in Supplementary Fig. 16 (reproduced below).

These results provide the feasibility for the integration of BSO with Si substrates. While the growth of BSO on CMOS-compatible substrates is definitely an interesting direction of research, it is beyond the focus of this work, and requires a more dedicated and separate study.

In summary, we focus on the prototype demonstration and analysis of the high TER, multilevel resistance states, low variation, and high endurance properties using the large polarization and high quality of the BSO film grown on NSTO substrate. While challenges remain on further increasing the speed, decreasing the energy, and fabricating CMOS-compatible devices, we believe that with

the progresses mentioned above and proper trade-off based on the specific application, the BSO-based FTJs are highly promising for nonvolatile memories and in-memory computing.

We have added the following text, figure, and references in the revised manuscript to help clarify this point, which we reproduce below:

“The NSTO substrate has been a common substrate for constructing high-performance FTJs^{6,19,44}. While the depletion region at the surface of NSTO may lead to a larger resistance compared with a metal contact, the high switching speed of 300 ps and low programming energy below 10 fJ have been demonstrated in FTJs using the NSTO substrate⁴². Furthermore, although currently the NSTO substrate instead of Si substrate is used, it is promising to obtain BSO films on substrates compatible with complementary–metal–oxide–semiconductor (CMOS) fabrication processes. For example, ultrathin SrTiO₃ or similar films can be grown on silicon-on-insulator (SOI) wafers using molecular beam epitaxy (MBE)^{48,49}, or bonded to the Si substrate through the ion slicing, or the “smart-cut” process^{50,51}, which could also be used for BSO film growth. Furthermore, by using a water-soluble Sr₃Al₂O₆ as the sacrificial layer, the SrTiO₃ and BiFeO₃ thin films have been released from the growth substrate in water, and transferred to a Si substrate. Such technique may also be used to transfer the BSO films. Besides, we demonstrate the initial results for the growth of the BSO film on metal-on-oxidized-Si substrates such as the Au/SiO₂/Si substrate (Supplementary Fig. 16). These results show high potential for the integration of BSO films onto Si substrates, and for on-chip memories using BSO-based FTJs.”

“

Figure S16. The XRD pattern of the BSO film grown on (111) Au/SiO₂/Si substrate.

”

References:

“

- 48 Ortmann, J. E. et al. Epitaxial oxides on glass: a platform for integrated oxide devices, *ACS Appl. Nano Mater.* **2**, 7713–7718 (2019).
- 49 Xiong, C. et al. Active silicon integrated nanophotonics: ferroelectric BaTiO₃ devices, *Nano Lett.* **14**, 1419–1425 (2014).
- 50 Roth, R. M. et al. Fabrication and material properties of submicrometer SrTiO₃ films exfoliated using crystal ion slicing, *Appl. Phys. Lett.* **90**, 112913 (2007).
- 51 Lee, Y. S. et al. Fabrication of patterned single-crystal SrTiO₃ thin films by ion slicing and anodic bonding, *Appl. Phys. Lett.* **89**, 122902 (2006).

”

Reviewer #1: “1) The authors need to show evidence that the memory switching characteristics in the device is not due to other type of transport mechanism such as by filamentary switching. Filamentary switching can be possible in oxide materials like the BSO. Such additional measurement that should be carried out include temperature dependent I-V characteristics”

We thank Reviewer #1 for this insightful comment. We agree with Reviewer #1 that it is crucial to show evidence that the memory switching characteristics are not due to filamentary switching, and we have indeed performed additional measurement and analysis to show the ferroelectric switching mechanism.

First of all, for filamentary switching, usually the conduction filament (CF) should initially be formed during the high-voltage “forming” process, and then the CF will be partially ruptured or connected during the reset or the set processes. The forming process usually needs a larger voltage than the set process, as shown in most of the filamentary-type resistive switching devices without special forming-free designs [Ref. 52: Traoré, B. et al. Microscopic understanding of the low resistance state retention in HfO₂ and HfAlO based RRAM. *International Electron Devices Meeting (IEDM)*, 546–549 (IEEE, San Francisco, CA, USA, 2014); Ref. 53: Yang, R. et al. Ternary content-addressable memory with MoS₂ transistors for massively parallel data search, *Nature Electronics* **2**, 108–114 (2019)]. However, in our devices, such a forming phenomenon is not observed.

Second, based on Reviewer #1's suggestion, we have performed temperature-dependent I - V measurements, as shown in Supplementary Fig. 17 in the revised manuscript (reproduced below). We find that the current increases with temperature in both HRS and LRS, with a larger current increase in HRS (Supplementary Fig. 17a and 17b). The trend in LRS is contrary to the filamentary switching. For filamentary switching, the CF in LRS will lead to metallic conduction, which means that the LRS resistance should increase and the current should decrease with higher temperature due to increased phonon scattering, as shown in the previous work [Ref. 54: Ling, Y. et al. Temperature-dependent accuracy analysis and resistance temperature correction in RRAM-based in-memory computing, *IEEE Electron Device Lett.* DOI: 10.1109/TED.2023.3266186 (2023)]. However, in our device, the LRS resistance slightly decreases with temperature, which suggests that direct tunnelling is the dominant conduction mechanism in LRS, with some contribution from thermally assisted tunnelling. Such conduction mechanism is consistent with ferroelectric switching in LRS, but not for filamentary switching.

Third, in HRS, the measured results can be well explained by thermally assisted tunnelling, which is consistent with ferroelectric switching in HRS. Fitting to the experimental data is performed using thermally assisted tunnelling model described in detail in Supplementary Note 1.2. The thermally assisted tunnelling current is given by: $I = A \cdot J_0 \exp(qV/E_0)$, where q is the electron charge, V is the applied voltage, and A is the device area. E_0 is a temperature-dependent energy given by: $E_0 = nk_B T$, where n is the ideality factor which equals to unity for ideal thermionic emission over a Schottky-type barrier, k_B is Boltzmann constant, and T is the temperature. The temperature-dependent saturation current J_0 is given by:

$$J_0(T) = \frac{A^* T^2 \pi^{1/2} E_{00}^{1/2} [q(V_{bi} - V) + \phi_n]^{1/2}}{k_B T \cosh(E_{00}/k_B T)} \times \exp\left(\frac{\phi_n}{k_B T} - \frac{qV_{bi} + \phi_n}{E_0}\right), \quad E_{00} = \frac{qh}{4\pi} \left(\frac{N_D}{m^* \varepsilon_r(T) \varepsilon_0}\right)^{1/2},$$

where A^* is the effective Richardson constant, V_{bi} is the built-in potential of the Schottky-like barrier due to the surface depletion, N_D is the doping concentration, $\varepsilon_r(T)$ is the temperature-dependent relative permittivity of NSTO, and ϕ_n is the difference between the conduction band minimum (E_C) and the Fermi level (E_F) of NSTO. Through fitting to the measured I - V characteristics, we extract the ideality factor n at different temperatures as shown in Supplementary Fig. 17c. We find that as the temperature decreases, n increases, which suggests that thermally assisted tunnelling becomes more significant, consistent with previous reports [Ref. 35: Cuellar, F.

A. et al. Thermally assisted tunneling transport in $\text{La}_{0.7}\text{Ca}_{0.3}\text{MnO}_3/\text{SrTiO}_3$: Nb Schottky-like heterojunctions. *Phys. Rev. B* **85**, 245122 (2012); Ref.: Ruotolo, A. et al. High-quality all-oxide Schottky junctions fabricated on heavily doped Nb:SrTiO₃ substrates. *Phys. Rev. B* **76**, 075122 (2007)]. Fitting to the slope in $\ln(J_0 \cosh(E_{00}/k_B T)/T)$ vs. $1/E_0$ plots allows the extraction of $-qV_{bi}$ (Supplementary Fig. 17 e). The energy barrier qV_{bi} is extracted to be 1.25 eV in HRS. In contrast, qV_{bi} of only about 0.1 eV is extracted in LRS, suggesting that direct tunnelling is indeed dominant in LRS with slight thermally activated conduction, while HRS is dominant by thermally assisted tunnelling, which is consistent with ferroelectric switching.

Fourth, we have performed additional density functional theory plus Hubbard U (DFT+ U) calculations, as shown in Fig. 3d, and Supplementary Figs. 7–9. The projected density of states (DOS) onto the Ti $3d$ orbitals for the interface layers show that the ferroelectric polarization modulates the DOS at the interface layers to change the resistance state. Our DFT+ U calculations also confirm the elimination of energy barrier in LRS and the existence of the extra barrier in HRS, which is consistent with the mechanism of ferroelectric switching.

Finally, we have also measured the C - V characteristics of the BSO-based FTJs in both LRS and HRS, as shown in Supplementary Fig. 18 (reproduced below). The lower capacitance in HRS suggests the existence of the Schottky barrier and thus the extra capacitance C_d in series with the capacitance from the ferroelectric layer C_f , so that the total capacitance C can then be obtained from $1/C = 1/C_d + 1/C_f = n/C_d$. As the HRS resistance increases, the Schottky barrier is enhanced, which has been discussed in detail in Supplementary Note 2. We find that C_d^{-2} increases almost linearly with a more negative reverse bias voltage, which can be explained by:

$$\left(\frac{1}{C_d}\right)^2 = \frac{2(V_{bi} - V_d)}{q\varepsilon_0\varepsilon_r N_D A^2}, \text{ where } V_d \text{ is the reverse-bias voltage that drops across the depletion region,}$$

and ε_0 is vacuum permittivity. Through the fitting, we extract the built-in potential qV_{bi} of 1.35 eV, which is comparable to that measured from the temperature-dependent I - V measurements. Therefore, the C - V measurement results are consistent with ferroelectric switching with an additional barrier due to the depletion region.

In summary, from the forming-free switching behavior, temperature-dependent LRS and HRS states, DFT+ U calculations, and C - V measurements, we find that the ferroelectric switching, instead of filamentary switching, is indeed dominant in these BSO-based FTJs.

We have added the following text, figure, and references in the revised manuscript to help clarify this point, which we reproduce below:

“

Figure S17. Temperature-dependent measurement and analysis of the 1-nm-thick BSO-based FTJ in both HRS and LRS. a–b, The measured I – V curves in **a** HRS and **b** LRS at different temperatures ranging from 100K to 300K, with the solid lines fitted by Supplementary equation (4). **c–d**, The extracted ideality factor n at different temperature in **c** HRS and **d** LRS. **e–f** The $\ln(J_0 \cosh(E_{00}/k_B T)/T)$ vs. $1/E_0$ plots for **e** HRS and **f** LRS, fitted by the Supplementary equation (6), from which V_{bi} of 1.25 V in HRS and 0.1 V in LRS are extracted.

”

“

Figure S18. The measured C - V properties of BSO-based FTJs. a, The capacitance of the BSO-based FTJ measured in LRS and HRS at different frequencies, showing that the depletion region form an extra capacitance in series with the ferroelectric capacitance in HRS. **b**, The measured C_d^{-2} - V_d characteristics (symbols) for the HRS of 1-nm-thick and 4.6-nm-thick BSO-based FTJs, with the solid lines showing the fitting to $C_d^{-2} = 2(V_{bi} - V_d) / q\epsilon_0\epsilon_r N_D A^2$. For FTJs with 1-nm-thick BSO, the V_{bi} is extracted to be 1.35 V, and the corresponding depletion width (W_d) is 9.72 nm. For FTJs with 4.6-nm-thick BSO, the V_{bi} is extracted as 2.04 V, and the corresponding $W_d = 11.8 \text{ nm}$.

”

“In oxide-based materials such as BSO, filamentary switching due to vacancy motion should also be considered. However, in these BSO-based FTJs, we do not observe the initial forming process that is required for most of the filamentary switching memories without special forming-free designs^{52,53}. We also measure the temperature-dependent I - V characteristics in both LRS and HRS (Supplementary Fig. 17), which shows that the current increases with temperature in both HRS and LRS, with a larger current increase in HRS. Such trend in LRS is contrary to the filamentary switching which should have a decreasing current with higher temperature due to the metallic conduction through the conductive filament⁵⁴. Through fitting, we find that direct tunnelling is dominant in LRS, and thermally assisted tunnelling is dominant in HRS, with the energy barrier of 0.1 eV extracted for LRS, and 1.25 eV extracted for HRS, consistent with ferroelectric switching (Supplementary Note 1.2). Furthermore, we measure the capacitances of the FTJs in LRS and HRS (Supplementary Fig. 18). The lower capacitance in HRS than that in LRS, and the capacitance-voltage relationship that fits well to the depletion capacitance model suggest the existence of a built-in potential and a depletion region (Supplementary Note 2). Therefore, ferroelectric switching should be the dominant mechanism in these BSO-based FTJs.”

References:

“

- 52 Traoré, B. et al. Microscopic understanding of the low resistance state retention in HfO₂ and HfAlO based RRAM. *International Electron Devices Meeting (IEDM)*, 546–549 (IEEE, San Francisco, CA, USA, 2014)
- 53 Yang, R. et al. Ternary content-addressable memory with MoS₂ transistors for massively parallel data search, *Nat. Electron.* **2**, 108–114 (2019).
- 54 Ling, Y. et al. Temperature-dependent accuracy analysis and resistance temperature correction in RRAM-based in-memory computing, *IEEE Electron Device Lett.* DOI: 10.1109/TED.2023.3266186 (2023).

”

Reviewer #1: “2) In Fig. 1, it seems that the coercive field is less than 2 V for the 1 nm BSO material. Why the reset voltage in the device is much higher (-4.5 V) for the FTJ device. The set and reset voltages are also significantly asymmetrical. The authors should clarify this.”

We thank Reviewer #1 for this insightful comment, which allows us to further clarify the mechanisms of FTJ switching. First of all, while we agree with Reviewer #1 that the coercive field is less than 2 V, we would like to clarify that the coercive field and reset voltage are different in definition. The coercive field indicates the voltage that the direction of polarization starts to switch, while the reset voltage is the maximum voltage used to reset the device, which can be higher than the voltage at the coercive field to ensure large enough polarization switching and a high TER. In fact, it can be observed from the resistance–voltage (R – V) measurement in Fig. 4f that the device starts to switch from LRS to HRS from ~ -1 V, which is consistent with the coercive field.

Second, the much higher voltage (-4.5 V) during the reset process is also related to the depletion region formed in the BSO/NSTO interface, which will divide a portion of the total voltage. The depletion region will only exist in the HRS, and will almost disappear when the polarization is switched so that the device is in LRS. Therefore, the set voltage is lower than the reset voltage, and is consistent with the coercive field. This also makes the set and reset curves asymmetric, which is a common phenomenon in various FTJs, as reported in literature [Ref. 13: Feng, N. et al. A physics-based dynamic compact model of ferroelectric tunnel junctions. *IEEE*

Electron Device Lett. **44**, 261-264 (2023); Ref. 19: Wen, Z. et al. Ferroelectric-field-effect-enhanced electroresistance in metal/ferroelectric/semiconductor tunnel junctions. *Nat. Mater.* **12**, 617–621 (2013)].

We have added the following text in the revised manuscript to make this point explicit:

“The resistance shows gradual change with set and reset voltages without abrupt switching, and the switching characteristics are highly repeatable (inset of Fig. 2a). We also find that a higher reset voltage than set voltage is needed, because of the extra depletion region formed at the BSO/NSTO interface during the reset process, which will divide a portion of the total voltage, as reported previously^{13,19}.”

Reviewer #1: “3) Fig. 5, the device used for endurance test in Fig 5a and 5b only has TER less than 1000. The author should use device with the claimed high TER for such tests. Also, the device used for the endurance test (Fig. 5a and 5b) and the device used for the retention test (Fig. 5c) has very different TER ratio and the data apparently comes from two different devices. It is important that for the 1 nm BSO thickness FTJ device, the I-V characteristics in Fig. 2a and 2b, the multi-level test in Fig. 4, the endurance and the retention test should be done on the same device. A good FTJ needs simultaneously have good features in these tests.”

We thank Reviewer #1 for raising this important issue. First of all, we agree with Reviewer #1 that “a good FTJ needs simultaneously have good features in these tests”, and we would like to clarify that indeed these measurements are performed on the same FTJ with 1-nm-thick BSO layer. The differences of the TER among different measurements are mainly because of the different programming conditions used, instead of using different devices. As we show in Fig. 4d-4f, the pulse number, pulse duration, and programming voltage can all change the TER of the same device. Specifically, during the measurement for this FTJ, we first measure the DC $I-V$ properties, followed by the analog properties and the retention, and finally we perform the endurance test because the device may degrade during the endurance measurements.

Second, we also agree with Reviewer #1 that the endurance test should be performed with a high TER. Indeed, as shown in Supplementary Fig. 10 during our initial submission (Supplementary Fig. 12 in the revised manuscript), we have already demonstrated endurance measurements with $TER > 10^5$ for the FTJ based on 1-nm BSO using the set pulse of 2.5 V and

reset pulse of -5 V. The device shows very slight degradation after $\sim 5 \times 10^4$ endurance cycles. To further explore the endurance properties on the same device, we performed measurements by using lower programming voltages of 2.2 V set pulse and -3.5 V reset pulse, which resulted in a TER over 300 and an endurance cycle over 10^9 . Such TER is already much larger than the TER of FTJs with 1-nm ferroelectric layers, and the endurance over 10^9 is useful for applications that requires a moderate TER but high endurance, such as neural network training or online learning [Ref.: Yu, S., Neuro-inspired computing with emerging nonvolatile memory, *Proc. IEEE* **106**, 260-285 (2018)]. From the measurement, we find a trade-off between the TER and the endurance cycle, where a higher TER usually leads to a lower endurance cycle or earlier degradation, presumably due to the larger voltage applied, as reported previously [Ref. 23: Cheema, et al. One nanometer HfO₂-based ferroelectric tunnel junctions on silicon. *Adv. Electron. Mater.* **8**, 2100499 (2022); Ref. 42: Luo, et al. High-precision and linear weight updates by subnanosecond pulses in ferroelectric tunnel junction for neuro-inspired computing. *Nat. Commun.* **13**, 699 (2022)]. Despite such tradeoff, our 1-nm-BSO-based FTJs show high endurance with relatively high TER, due to the high quality and high polarization of the BSO layer.

Third, the properties among different devices are quite consistent. As shown by the measurement results from 35 different devices in Fig. 5d in the Main Text, the $\text{TER} > 10^5$ has been consistently observed. Furthermore, multilevel conductance tuning properties among different devices have also been demonstrated in Supplementary Fig. 10.

Finally, we have further performed measurements on the endurance properties with the high TER on another 1-nm-BSO-based FTJ until the device fails (the previous FTJ in Fig. 5a has degraded after the endurance measurement), as shown in Supplementary Fig. 14 (reproduced below). We measure the endurance properties using high voltages, and observe TER over 10^5 for up to 10^6 cycles. Due to the higher voltage than that in Fig. 5a in the Main Text, we observe that the memory window degrades faster than that in Fig. 5a. Nevertheless, we find that the FTJ can sustain a TER over 10^2 for more than 10^9 endurance cycles, and can remain alive after 5×10^9 cycles although with a lower TER. Such endurance cycle and TER are much better than the previously reported FTJs at such ferroelectric layer thickness.

To make this point even more clear, we have added the following figure and text in the revised manuscript, which we reproduce below:

“We measure the endurance properties by first applying high programming voltage pulses of 2.5 V for set and -5 V for reset, and show TER over 10^5 for about 5×10^4 endurance cycles (Supplementary Fig. 12). To achieve even higher endurance, we slightly decrease the programming voltages to 2.2 V for set and -3.5 V for reset, and demonstrate endurance up to 10^9 cycles with no significant degradation of TER (TER over 300). In addition, the total switching endurance up to 5×10^9 with TER ~ 100 for the 1-nm-BSO-based FTJ is demonstrated (Fig. 5a). While the TER degrades after 5×10^9 cycles, the I - V sweeps measured during the endurance switching cycles prove that the device is still functional (Supplementary Fig. 13), and the resistance window can be recovered by slightly increasing the voltage amplitude (Fig. 5b).”

“Furthermore, we measure the endurance of another FTJ based on 1-nm BSO, using 2.5 V set pulses and -5 V reset pulses until the device degrades (Supplementary Fig. 14). We find that the TER remains above 2×10^5 for over 10^5 cycles, and then starts to degrade. Nevertheless, the total endurance cycle over 5×10^9 cycles is still demonstrated.”

“

Figure S14. The endurance measurement of 1-nm-BSO-based FTJs with +2.5 V and -5 V programming pulse voltages. **a**, The endurance measured for 5×10^9 cycles. The device can sustain a TER over 10^5 for more than 10^6 cycles, with a total endurance cycle over 5×10^9 . **b**, The I - V switching characteristics measured during the endurance test. The device performance slightly degrades after 10^7 endurance cycles, and the switching window further shrinks as the endurance cycle further increases.

”

Reviewer #1: “4) The coercive field and remnant polarization of the BSO material used in this work needs to be stated in the manuscript.”

We thank Reviewer #1 for the constructive comment. We have summarized the coercive field and remanent polarization of the BSO film used in this work shown in Supplementary Table 1, and added the relevant text, which we reproduced below.

“

Table S1. The coercive fields and remanent polarizations of BSO films with different thicknesses.

Thickness (nm)	Coercive Field (MV/mm)	Remanent Polarization($\mu\text{C}/\text{cm}^2$)
1	1	16.6
2.1	0.95	38
4.6	0.75	50

”

“For BSO films with different thicknesses, the coercive fields and remanent polarizations are summarized in Supplementary Table 1.”

Reviewer #1: “5) Fig. 1c, the TEM image seems to show a transition region at the interface of BSO and Au/Cr metal (or the BSO layer seems to be thicker than 1 nm). The authors need to clarify this.”

We thank Reviewer #1 for raising this important issue. First of all, we would like to clarify that although the calculated size of the BSO unit cell is ~ 0.9 nm, some regions in the 1-nm film after growth may have three Bi-O layers, while some regions have four Bi-O layers. In regions with three Bi-O layers, the thickness is ~ 0.9 nm; and in regions with four Bi-O layers, the thickness can reach ~ 1.1 nm. The STEM image presented in Fig. 1c in the Main Text is four layers, so it is slightly thicker than 1 nm. To explain the structure of the BSO film with thickness of 1 nm more clearly, we present the three-dimensional atomic model from theoretical calculation (Supplementary Fig. 1). Furthermore, due to the stress of the substrate and surface, the lattice of the ultrathin BSO film along the c axis is stretched to a certain extent, which results in a stable polarization and makes the film appear slightly thicker. Finally, X-ray reflectivity measurements confirm that the thickness of the film is 1 nm with a surface roughness of 0.173 nm, which is consistent with our TEM image and theoretical model.

To make this point more explicit, we have added the following figure and text in the revised manuscript, which we reproduce below:

“The atomic model of the BSO thin film is shown in Supplementary Fig. 1, and the atomic arrangement of the BSO film with thickness down to ~ 1 nm is confirmed by the high-angle annular dark-field scanning transmission electron microscopy (HAADF-STEM) (Fig. 1c).”

“At the same time, X-ray reflectivity measurements and the fitting results confirm that the thickness of the BSO film is ~ 1 nm with a surface roughness of 0.173 nm^{24} , which is consistent with the theoretical model and the TEM image.”

“

Figure S1. The atomic model of BSO thin film, with the dashed blue box showing the BSO film illustrated in Fig. 1b. In the grown BSO film, some regions have three Bi-O layers, while some regions may have four Bi-O layers. In regions with three Bi-O layers, the thickness is ~ 0.9 nm; and in regions with four Bi-O layers, the thickness can reach ~ 1.1 nm. The X-ray reflectivity measurements and the fitting results confirm that the thickness of the BSO film is 1 nm with a surface roughness of 0.173 nm.

”

(II) Response to Reviewer #2

Please find below our point-by-point response to Reviewer #2 below. The original comments are in blue color and Calibri font; responses are in black Times New Rome.

Reviewer #2: "In the present manuscript, the authors designed FTJs based on Sm-substituted layered bismuth oxide (BSO). They demonstrated that these FTJs can achieve a TER over 7×10^5 with a 1-nm ferroelectric BSO film and three orders of magnitude higher than previous reports with such thickness. They also showed that the TER can be further enhanced as the thickness of the BSO layer increases and these FTJs demonstrate up to 32 resistance states. The experimental data are interesting and exciting. However, I think the theoretical explanation is not robust enough to support the present experimental findings. Therefore, I do not think the current version of the manuscript is suitable for publication in Nature Communications. Below are the reasons."

We thank Reviewer #2 for rating our experimental data as "interesting and exciting". We also thank Reviewer #2 for all the constructive comments and suggestions on the theoretical explanation, which are very helpful for improving our manuscript. We are very pleased to provide more computational proof to justify our experimental discovery.

Reviewer #2: "1) In the DFT calculation, the authors employed a supercell structure composed one-unit cell of ferroelectric bismuth oxide and four layers of NSTO to study the electronic structure of the interface between the ferroelectric layer and NSTO. Please provide a detailed interface atomic structure model."

We thank Reviewer #2 for raising this important issue. We have added the relevant text in the Main Text and the figure showing the interface supercell in the Supporting Information (Supplementary Fig. 7), which we reproduce below.

"The interface model can be found in Supplementary Fig. 7."

"

Figure S7. The supercell structure of the interface between bismuth oxide and NSTO. a, ‘up’ polarization; b, ‘down’ polarization. The polarization switching is simulated by rotating the structure of ferroelectric layer by 180 degrees. The NSTO has a TiO₂ termination that is energetically favored. The dashed line indicates the interface between the two different components. To reduce the computational difficulty, Sm is not included in this model. It should be mentioned that the ferroelectricity mainly comes from the bismuth atoms, and the role of Sm is not crucial to the interfacial states (evidenced by Supplementary Fig. 8).

”

Reviewer #2: “2) The authors used the DFT+*U* method, which did not consider the *U* value of Sm atom, but only considered Ti atom. The existence of f electrons in Sm atoms is difficult to converge, and the *U* value needs to be considered.”

We thank Reviewer #2 for the insightful comment. As evidenced by our former DFT and ab initio molecular dynamics study [Yang, Q. et al. *Science* **379**, 1218–1224 (2023)], although Sm plays a significant role in maintaining the thermodynamics stability, the ferroelectricity in BSO layers results from the lone pair electrons of Bi. In the experiment, both the ferroelectric bismuth oxides and the NSTO substrates are prepared in the form of doped material. However, such complexity introduces tremendous difficulty in the DFT simulations. To reduce the simulation complexity of the interface, we have used BO/NSTO models instead of the actual BSO/NSTO interface. This approach is reasonable because: 1) the ferroelectricity mainly comes from bismuth atoms, 2) DFT calculations are performed at 0 K, and our former study [Yang, Q. et al. *Science* **379**, 1218–1224 (2023)] has suggested that the polarization in BO can be maintained very well at 0 K, 3) As evidenced by the hybrid DFT HSE06 calculations of the bulk Sm-doped bismuth oxide, the

electronic states near the Fermi level are much weaker than the other electronic states of Bi and O (Fig. R2), and thus its effect on the crucial electronic properties should be very small.

Figure R2: The DOS calculated by hybrid DFT HSE06 calculations for Sm substituted Bi_6O_9 in bulk form. The Fermi energy is set to zero.

To confirm the negligible role of Sm in the electronic properties at the interface, we have performed additional calculations to gain further insights into this issue. Specifically, in our added calculations, we have simulated an interface BSO/NSTO model with “up” polarization of BSO, in which one Bi atom is replaced by one Sm atom. The crystal structure is shown in Supplementary Fig. 8a (reproduced below). In line with a recent study of Sm-based compounds [Ref.: Banerjee, D. et al. Pressure-induced electronic transitions in samarium monochalcogenides. *Phys. Rev. B* **105**, 195135 (2022)], we have used the Hubbard U of 6 eV and J of 0.855 eV for Sm. The other computational details are the same as those as shown in the Main Text. The interfacial density of states (DOS) of Ti are shown in Supplementary Fig. 8b. Compared with Fig. 3d in the Main Text, we don’t find significant difference, implying that the effect of the Sm on the interfacial states can be neglected. We have also compared the element density of states of Sm with the total DOS of the whole interface structure (Supplementary Fig. 8c), and have found that the states near the Fermi level are very weak, which is consistent with the above results of Sm-doped Bi_6O_9 in bulk form calculated by the HSE06 (Fig. R2). In addition, the dominating electronic states of Sm lie as deep

as around -20 eV. Therefore, our calculations imply that even if we exclude the Sm doping, it will not strongly affect the electronic properties at the interface.

To justify the use of our simplified model, we have added the following figure and text in the revised manuscript:

"We further demonstrate that the effect of Sm atoms on the interfacial DOS and electronic properties at the interface is negligible (Supplementary Fig. 8)."

"

Figure S8: The DFT results of the BSO/NSTO interface. **a.** Interface structure of BSO/NSTO. **b.** Interface states of Ti, calculated using the Hubbard U of 6 eV and J of 0.855 eV for Sm. **c.** The comparison between Sm DOS and total DOS, showing that the states near the Fermi level are very weak, and the dominating electronic states of Sm lie as deep as around -20 eV. Therefore, excluding the Sm will not strongly affect the electronic properties at the interface.

Reviewer #2: "3) How do theoretical calculations simulate the flipping of the ferroelectric polarization direction of BSO? Is it just to rotate the BSO 180 degrees? Does it match the experiment? Please give a detailed explanation."

We thank Reviewer #2 for giving us the opportunity to clarify this issue. To simulate 180-degree polarization switching, we rotated the bismuth oxide cell accordingly, as schematically depicted in the interface structures (Supplementary Fig. 7, shown above). Based on the experimental PFM measurement in Fig. 1e in the Main Text, we have confirmed the 180-degree polarization switching. In our computations, our primary focus is on simulating the ferroelectric switching effect on the interface properties. Therefore, we rotated the structure to model the polarization switching in bismuth oxide and examined the corresponding interface properties across different polarization states.

To help further clarify this point, we have included the following text in the caption of Supplementary Fig. 7:

"The polarization switching is simulated by rotating the structure of ferroelectric layer by 180 degrees."

Reviewer #2: "4) The authors only provided the electronic properties of the BSO-NSTO interface in flipping the ferroelectric polarization direction, how about the BSO-Cr/Au interface? Perhaps differential charge densities would provide more robust evidence. Please give the corresponding result."

We thank Reviewer #2 for the insightful comment. We agree with Reviewer #2 that the study of properties at the interface of the metal electrodes and BSO may also provide useful information to understand the experimental results. In analogy to the simulation of the interface between the ferroelectric layer and the NSTO, we have also used the BO layer to reduce the computational complexity. In the experiment, because Cr is relatively thick (15 nm), we can eliminate the possible interactions between the Au and the ferroelectric layers. Therefore, we have only included Cr (4 layers of unit cells) in the electrode in the simulation models, leading to BO/Cr slab models with vacuum layer exceeding 15 angstroms.

We show the charge density difference as the polarization is oriented up and down in Supplementary Fig. 9. It is apparent that there is charge transfer at the interface, and this charge transfer can be tuned by the polarization switching. It is found that the bottom atomic layer of bismuth oxide gets some charge at the interface in both cases, and the charge transfer in the "up"

state is stronger than the “down” state. Such different charge density states indicate that the BO/Cr interface may also help in achieving the giant tunnelling electroresistance.

To make this point more explicit, we have added the following figure and text in the revised manuscript:

“In addition, we have simulated the interface between bismuth oxide and Cr. The charge density difference in Supplementary Fig. 9 shows that the charge transfer can be modulated by the ferroelectric switching, which indicates that the Cr/BSO interface could also contribute to the large TER.”

“

Figure S9: Charge density difference at the interface between the ferroelectric bismuth oxide and metallic chromium layers, with the bismuth oxide layer showing (a) upward, and (b) downward polarizations.

”

Reviewer #2: “5) In Figure 3c and Figure S5, the I-V curve fitted by the direct tunneling model is extremely perfect with the experimental results, which makes one have to doubt the authenticity of the theoretical fitting data. Can a more accurate theoretical calculation be given based on the non-equilibrium Green's function method?”

We thank Reviewer #2 for the constructive comment. First of all, we would like to clarify that when fitting to the experimental I - V results, instead of directly calculating the results using the first principles calculation, we fit the experimental data using a phenomenological model with fitting parameters, which ensures nice agreement between the experimental results and the model fitting. For example, in the model, the thermally assisted tunnelling current is given by: $I = A \cdot J_0 \exp(qV/E_0)$, where q is the electron charge, V is the applied voltage, J_0 is the reverse saturation current, and A is the device area. E_0 is a temperature-dependent energy given by: $E_0 = nk_B T$, where n is the ideality factor which equals to unity for ideal thermionic emission over a Schottky-type barrier, k_B is Boltzmann constant, and T is the temperature. When fitting to the thermally assisted tunnelling model, the ideality factor n will determine the slope and J_0 will determine the current level. Therefore, the parameters n and J_0 can be extracted from fitting to the experimental data, as shown in Fig. 3c in the Main text. Besides, based on the extracted n and J_0 at different temperatures, we obtain the $\ln(J_0 \cosh(E_{00}/k_B T)/T)$ vs. $1/E_0$ plot, and extract the built-in potential $-qV_{bi}$ from the slope, as shown in Supplementary Fig. 17.

Second, the direct tunnelling and thermally assisted tunnelling models in FTJs have been discussed and verified in previous works with the similar fitting method, where they also show good agreement between the experimental data and the fitting results, such as in [Ref. 35: Cuellar, et al., Thermally assisted tunneling transport in $\text{La}_{0.7}\text{Ca}_{0.3}\text{MnO}_3/\text{SrTiO}_3$: Nb Schottky-like heterojunctions. *Phys. Rev. B* **85**, 245122 (2012), and Ref. 20: Xi, et al., Giant tunnelling electroresistance in metal/ferroelectric/semiconductor tunnel junctions by engineering the Schottky barrier. *Nat. Commun.* **8**, 15217 (2017)]. Therefore, the nice agreement between the experiment data in our device and the model proves that the thermally assisted tunnelling indeed dominates in HRS, and direct tunnelling dominates in LRS.

Third, as for the non-equilibrium Green's function method, we fully agree with Reviewer #2 that non-equilibrium Green's function method may give more accurate modeling results, which may supplement our fitting results based on the tunnelling models, because it can compute the conductance of the ferroelectric tunnel junction and the respective transmission spectra for each polarization state. However, because the tunnelling film is as thin as only 1 nm with dopants, and both the ferroelectric tunnelling barrier (bismuth oxide) and the electrode are prepared in a doped form (*i.e.*, Nb doped STO, and Cr/Au electrodes), the theoretical calculation based on non-

equilibrium Green's function method will be very challenging due to the existence of several dopants. Indeed, even with the standard DFT calculations, we have encountered many difficulties, which eventually results in introducing many approximations into the model, such as the excluding of Sm and Au in the DFT simulations. Considering the necessary approximations involved in density functional non-equilibrium Green's function calculations, it is unlikely that these more complex computations would provide a superior interpretation of the experimental results compared to the tunnelling models employed in our manuscript, since the fitting to experimental data and the extraction of the parameters have been well achieved using the tunnelling models as Reviewer #2 has commented. The computationally intensive analyses may be better suited for individual theoretical endeavors in the future.

Finally, although the exact Green's function modeling results based on our actual ferroelectric tunnel junction is challenging, we notice that Shen et al. have proposed the theoretical model of a 2D-FTJ by utilizing a much approximated model of In:SnSe/SnSe/Sb:SnSe homostructure that is very similar to our situation because doped semiconductors have been used as electrodes [Supplementary Ref. 6: Shen, et al. Two-dimensional ferroelectric tunnel junction: the case of monolayer In:SnSe/SnSe/Sb:SnSe homostructure. *ACS Appl. Electron. Mater.* **1**, 1133–1140 (2019)]. They have utilized density functional non-equilibrium Green's function methods and observed that the device density of states and the double well of the ferroelectric tunnel junction indicate that both barrier width and barrier height are modulated by ferroelectric switching, resulting in distinct quantum tunnelling conductance behaviors. The mechanisms revealed in the study by Shen et al. should also work in our case.

To stimulate future endeavors from ourselves and other researchers in the community, we have added the following text and reference in the Supplementary Information:

“(1.3) Discussion on theoretical investigations of transport properties

Although the tunnelling mechanism in the ferroelectric tunnel junctions employing doped semiconducting electrodes as in this work has been elucidated in a former study by Shen et al⁶, further theoretical investigations based on density functional non-equilibrium Green's function methods are worth carrying out to supplement the tunnelling model for fully understanding the transport properties of our studied ferroelectric tunnel junctions.”

Supplementary Reference:

“[6]. Shen, X-W. et al. Two-dimensional ferroelectric tunnel junction: the case of monolayer In:SnSe/SnSe/Sb:SnSe homostructure. *ACS Appl. Electron. Mater.* **1**, 1133–1140 (2019).”

Reviewer #2: “(6) The colors of Sm atoms and Bi atoms in the atomic structure model in Fig.1 (b) are difficult to distinguish, please modify.”

We thank Reviewer #2 for the constructive comment. We agree with Reviewer #2 that the colors of Sm atoms and Bi atoms are similar in Fig. 1b. Therefore, we have modified the color of Sm atoms in revised Fig. 1b to distinguish from Bi atoms, as shown in Fig. R3.

Figure R3. Atomic structure of BSO. Structure in (a) the manuscript during the initial submission, and (b) the revised manuscript.

To make this point explicit, we have changed Fig. 1b in the Main Text, which we reproduce below:

“

”

Reviewer #2: “7) Tunneling or Tunnelling, Programming or Programing? Please use consistent spelling throughout the manuscript.”

We thank Reviewer #2 for raising the important issue. We have made necessary modifications to ensure that consistent spellings are used throughout the manuscript, including “tunnelling”, “programming”, and other wording.

(III) Response to Reviewer #3

Please find below our point-by-point response to Reviewer #3 below. The original comments are in red color and Calibri font; responses are in black Times New Rome.

Reviewer #3: “Jia et al. report on the novel BSO-based ferroelectric tunnel junctions that can maintain a large TER over 10^5 even with the film thickness down to 1 nm, three orders of magnitude higher than previous results at similar film thickness. Previously, the scaling down of FTJ is highly limited by the critical-thickness effect, because the polarization is shielded as the thickness decreases due to the huge depolarization field. Therefore, the demonstration of such large TER ratio at such small thickness makes a significant advance to the field, and is meaningful to the community. The reliable multi-level cells and analog properties also make these devices promising for various applications. The manuscript is well organized, with solid experimental results and theoretical analysis. After the following issues are further explained and addressed, the manuscript should be suitable for publication in Nature Communications.”

We thank Reviewer #3 for rating that our work “makes a significant advance to the field”, and is “meaningful to the community”. We also thank Reviewer #3 for the comment that our manuscript is “well organized”, and the experimental results and theoretical analysis are “solid”. We are very glad to address the problems raised by the reviewer.

Reviewer #3: “(1) In Fig. 1b, it seems that the Bi/Sm atoms are not aligned, and the Sm atoms only appear in the upper half of the crystal structure, while there is no explanation about the structure. Besides, the thickness of the BSO layer shown in Fig. 1c seems to be slightly larger than 1 nm. Why it is not consistent with the X-ray reflectivity measurement results?”

We thank Reviewer #3 for this insightful comment. To explain the structure of the BSO film with thickness of 1 nm more clearly, we present the three-dimensional atomic model from theoretical calculation (Supplementary Fig. 1), in which the position of Sm is set in the Bi-O(3) layer. In the FTJ structure shown in Fig. 1b in the Main Text, the interface between the 1-nm BSO film and the substrate is connected by Bi-O(2) and Ti-O, and the presented BSO layer is successively Bi-O(2) — Bi-O(1) — Bi-O(3) — Bi-O(2) from bottom to top (dashed blue box in Supplementary Fig. 1). In fact, the Sm atoms are distributed randomly in the structure. In the manuscript, we provide one

possible BSO structure obtained from our previous study [Ref. 24: Yang, Q. *et al. Science* **379**, 1218-1224 (2023)]. Furthermore, the unaligned atoms are based on theoretical calculation results with the consideration of the formation energy, band gap, and ferroelectricity of the structure, which is consistent with the structure shown in Ref. 24.

Besides, while the calculated size of the BSO unit cell is about 0.9 nm, some regions in the 1-nm film after growth may be three layers of Bi-O, while some regions are four layers of Bi-O. In regions with three Bi-O layers, the thickness is 0.9 nm; and in regions with four Bi-O layers, the thickness can reach 1.1 nm. The STEM image presented in Fig. 1c in the Main Text is four layers, so it is slightly thicker than 1 nm. Furthermore, due to the stress of the substrate and surface, the lattice of the ultrathin BSO film along the *c* axis is stretched to a certain extent, which results in a stable polarization and makes the film appear slightly thicker. Finally, X-ray reflectivity measurements confirm that the thickness of the film is 1 nm with a surface roughness of 0.173 nm, which is consistent with our TEM image and theoretical calculation.

To make this point more clear, we have added the following figure and text in the revised manuscript, which we reproduce below:

"While the Sm atoms are distributed randomly in the structure, we provide one possible BSO structure obtained from our former study²⁴."

"The atomic model of the BSO thin film is shown in Supplementary Fig. 1, and the atomic arrangement of the BSO film with thickness down to ~1 nm is confirmed by the high-angle annular dark-field scanning transmission electron microscopy (HAADF-STEM) (Fig. 1c)."

"At the same time, X-ray reflectivity measurements and the fitting results confirm that the thickness of the BSO film is ~1 nm with a surface roughness of 0.173 nm²⁴, which is consistent with the theoretical model and the TEM image."

"

Figure S1. The atomic model of BSO thin film, with the dashed blue box showing the BSO film illustrated in Fig. 1b. In the grown BSO film, some regions have three Bi-O layers, while some regions may have four Bi-O layers. In regions with three Bi-O layers, the thickness is ~ 0.9 nm; and in regions with four Bi-O layers, the thickness can reach ~ 1.1 nm. The X-ray reflectivity measurements and the fitting results confirm that the thickness of the BSO film is 1 nm with a surface roughness of 0.173 nm.

”

Reviewer #3: “(2) In the local PFM measurement shown in Fig. 1f, the phase change is $\sim 120^\circ$ instead of the typical 180° , and the saturation in phase is observed. Is there any explanation on this effect?”

We thank Reviewer #3 for this insightful comment. We agree with Reviewer #3 that the phase change in PFM measurement is $\sim 120^\circ$. This is because although the Sm bonds help stabilize the ferroelectricity, the influence of depolarization field still increases as the thickness of the ferroelectric film decreases, which makes the ferroelectric domain switching more difficult, and the domain switching could be incomplete at several nanometers of thickness. This phenomenon of decreased phase change with smaller thickness has also been reported previously, such as the strain-free freestanding 2D SrTiO₃ film [Ref.: Zhang Y, et al. Room-temperature electric field-induced out-of-plane ferroelectric polarization in strain-free freestanding 2D SrTiO₃ membranes. *APL Mater.*, **11**, 041103 (2023); Ref. 26: Ji, D. et al. Freestanding crystalline oxide perovskites down to the monolayer limit. *Nature* **570**, 87–90 (2019)]. Nevertheless, we demonstrate that the ferroelectric polarization in BSO film, and the TER values in nanometer-thick BSO-based FTJs show better performance than previously reported FTJs with similar ferroelectric layer thickness.

Reviewer #3: “(3) The authors use STO substrate with 0.7 wt% Nb doping. Why is such doping concentration used, and have the authors considered using substrates with other doping

concentrations?”

We thank Reviewer #3 for the insightful comment, which provides us with the opportunity to further clarify this point. In our device, the selection of the NSTO substrate is based on two considerations. First of all, the semiconducting substrate allows the effective modulation of barrier width for higher HRS resistance, so the doping concentration should be moderate to allow the formation of depletion region, instead of being metallic. Second, a relatively high doping concentration should be used for decreasing the resistivity in LRS for a higher TER, and decreasing the voltage drop due to the NSTO resistance. Therefore, STO substrate with 0.7 wt% Nb doping is chosen to satisfy both requirements, and can lead to the giant TER in our devices. Furthermore, the concentration effect of Nb doping on the TER has been discussed in the previous work [Ref. 20: Xi, et al., Giant tunnelling electroresistance in metal/ferroelectric/semiconductor tunnel junctions by engineering the Schottky barrier. *Nat. Commun.* **8**, 15217 (2017)], which provides important guidance for the study in this work.

To make this point more clear, we have added the text in the corresponding position:

“The semiconductor substrate is STO with 0.7 wt% Nb doping, which provides low resistivity for decreasing the LRS resistance, and allows the efficient modulation of surface potential and the formation of the depletion region using the applied voltage or the polarization in the ferroelectric layer, as reported previously²⁰.”

Reviewer #3: “(4) In P.13, Line 230, the authors claim “32 distinct resistance states (5 bits of data storage) are achieved without any write-verification or write termination processes”. While 32 MLC achieved by simply varying reset voltages is remarkable, in Fig. 4b there seems to be some overlap among the neighboring resistance levels with variation considered. Will this affect the accuracy for memory/computing application, and are there any solutions that can avoid this issue?”

We thank Reviewer #3 for rating that the 32 MLC is “remarkable”, and for raising the important issue about resistance distribution, which provides us the opportunity to explain the proposed measurement technique in more details. We would like to clarify that we do not observe the overlap between most of the neighboring resistance states, and for certain neighboring resistance states, only some small overlap is observed when considering the worst-case cycle-to-cycle

variations. In many in-memory computing applications such as neural network inference, such level of device variation is typically tolerable, as demonstrated in [Ref. 42: Luo, et al. High-precision and linear weight updates by subnanosecond pulses in ferroelectric tunnel junction for neuro-inspired computing. *Nat. Commun.* **13**, 699 (2022)]. For the applications when even more accurate resistance states are required, our device can provide at least 16 resistance levels without any overlap, corresponding to 4 bits of memory. Besides, the variations can be further reduced and more resistance states without overlap can be achieved with extra programming efforts such as write-verification or write-termination methods.

To make this point more clear, we have added the following text in the revised manuscript:

“While some resistance overlap exists between certain neighboring resistance states when considering the worst-case cycle-to-cycle variation, such small variation is tolerable for many in-memory computing applications⁴². When even higher accuracy of multi-level storage is necessary, reducing the resistance levels to 16 (4 bits of memory) can avoid resistance overlap even for the worst-case cycle-to-cycle variation.”

Reviewer #3: “(5) The authors claim that they explore the linear tuning of multiple conductance states using two techniques: identical pulses and ISPP. It seems obvious from Fig.4c-d that the results obtained by ISPP method are much more linear than those obtained by the identical pulse method. Is the identical pulse method still useful for programming these devices then?”

We thank Reviewer #3 for the insightful comment. Although the result obtained by identical pulses is slightly more nonlinear compared with that obtained by the ISPP, it also has some advantages: it can save some peripheral circuits to control the voltage pulses, and the time delay and energy consumption for programming are smaller than the ISPP. Therefore, when linear conductance modulation is more desirable for certain applications, the ISPP programming technique should be used; and when the energy and delay are more important, the single identical pulse programming technique could be useful.

To make this point more clear, we have added the following text in the revised manuscript:

“While the ISPP programming technique is more suitable for linear conductance modulation, the identical pulse technique can save the additional control circuits and energy consumption. Therefore, it could be preferred for resource-constrained systems.”

Reviewer #3: "(6) In Fig. 5a-b, the TER clearly degrades after 5×10^9 endurance cycles. What is the physical mechanism of the degraded TER? What are the factors that can affect the endurance cycle in these FTJs? These should be more clearly explained."

We thank Reviewer #3 for this insightful comment, which helps us better explain the endurance properties of our device. During the repetitive switching, oxygen vacancies may gradually aggregate at the interface, which will lead to the ferroelectric "dead layer" [Ref. 45: Yang, Y. et al. Atomic-scale fatigue mechanism of ferroelectric tunnel junctions. *Sci. Adv.* **7**, eabh2716 (2021)]. The "dead layer" will then limit the domain switching and result in the degraded TER.

During the programming process, the applied voltage will highly affect the endurance cycles in the FTJs. A higher programming voltage will speed up the aggregation process, and will make the FTJ degrade faster. So a smaller voltage can be used to achieve a long endurance cycle, but with a slightly smaller TER. As shown in Supplementary Fig. 10 during our initial submission (Supplementary Fig. 12 in the revised manuscript), we have demonstrated endurance measurements with $TER > 10^5$ for the FTJ based on 1-nm BSO using set pulse of 2.5 V and reset pulse of -5 V. The device shows very slight degradation after $\sim 5 \times 10^4$ endurance cycles. Then we perform additional endurance measurement on the device by using lower programming voltages of 2.2 V for set pulses and -3.5 V for reset pulses, which results in a $TER \sim 300$ for the endurance cycle over 10^9 . Such tradeoff between the TER and the endurance cycle has also been reported previously [Ref. 23: Cheema, et al. One nanometer HfO₂-based ferroelectric tunnel junctions on silicon. *Adv. Electron. Mater.* **8**, 2100499 (2022); Ref. 42: Luo, et al. High-precision and linear weight updates by subnanosecond pulses in ferroelectric tunnel junction for neuro-inspired computing. *Nat. Commun.* **13**, 699 (2022)]. Despite such tradeoff, our 1-nm-BSO-based FTJs show high endurance with relatively high TER, due to the high quality and high polarization of the BSO layer.

In order to make this point explicit, we have added the following text in the revised Main text:

"The degraded TER could be attributed to the "dead layer" formed at the interface due to oxygen vacancy aggregation during the repetitively switching, which will limit the domain switching⁴⁵."

Reviewer #3: "(7) Some important references about ferroelectric switching are missing (e.g.

“Intrinsic ferroelectric switching from first principles”, *Nature* 2016; “Emergence of room-temperature ferroelectricity at reduced dimensions”, *Science* 2015).”

We thank Reviewer #3 for providing these references, and we have added them in the revised manuscript, as reproduced below.

“

[5] Lee, D. et al. Emergence of room-temperature ferroelectricity at reduced dimensions. *Science* **349**, 1314–1317 (2015).

[14] Liu, S., Grinberg, I. & Rappe, A. Intrinsic ferroelectric switching from first principles. *Nature* **534**, 360–363 (2016).

”

REVIEWERS' COMMENTS

Reviewer #2 (Remarks to the Author):

The authors have addressed my questions as much as possible and made corresponding revisions. The present version is acceptable for publication in Nature Communications.

Reviewer #3 (Remarks to the Author):

The authors have answered my concerns.

Response to Reviewers' Comments – NCOMMS-23-26395A

(I) Response to Reviewer #2

Please find below our point-by-point response to Reviewer #2 below. The original comments are in **blue color and Calibri font**; responses are in black Times New Rome.

Reviewer #2: “The authors have addressed my questions as much as possible and made corresponding revisions. The present version is acceptable for publication in Nature Communications.”

We thank Reviewer #2 for confirming that we have “addressed my questions as much as possible and made corresponding revisions”. We sincerely appreciate all the constructive comments from Reviewer #2, which have greatly enhanced our manuscript so that it “is acceptable for publication in Nature Communications”.

(II) Response to Reviewer #3

Please find below our point-by-point response to Reviewer #3 below. The original comments are in **red color and Calibri font**; responses are in black Times New Rome.

Reviewer #3: “The authors have answered my concerns.”

We thank Reviewer #3 for confirming that we have “answered concerns”. We sincerely appreciate all the constructive comments and important suggestions from Reviewer #3, which we believe have greatly enhanced our manuscript.